# Single-cell sequencing shows cellular heterogeneity of cutaneous lesions in lupus erythematosus

Meiling Zheng[1,2,6], Zhi Hu[1,2,6], Xiaole Mei[3,6], Lianlian Ouyang[1,2,6], Yang Song[1,2,6], Wenhui Zhou[1,2], Yi Kong[1,2], Ruifang Wu[1,2], Shijia Rao[1,2], Hai Long[1,2], Wei Shi[4], Hui Jing[1,2], Shuang Lu[1,2], Haijing Wu[1,2], Sujie Jia[5], Qianjin Lu ®[1,2,3] ✉ & Ming Zhao ®[1,2] ✉

Discoid lupus erythematosus (DLE) and systemic lupus erythematosus (SLE) are both types of lupus, yet the characteristics, and differences between them are not fully understood. Here we show single-cell RNA sequencing data of cutaneous lesions from DLE and SLE patients and skin tissues from healthy controls (HCs). We find significantly higher proportions of T cells, B cells and NK cells in DLE than in SLE. Expanded CCL20[+] keratinocyte, CXCL1[+] fibroblast, ISG[hi]CD4/CD8 T cell, ISG[hi] plasma cell, pDC, and NK subclusters are identified in DLE and SLE compared to HC. In addition, we observe higher cell communication scores between cell types such as fibroblasts and macrophage/dendritic cells in cutaneous lesions of DLE and SLE compared to HC. In summary, we clarify the heterogeneous characteristics in cutaneous lesions between DLE and SLE, and discover some specific cell subtypes and ligand-receptor pairs that indicate possible therapeutic targets of lupus erythematosus.

Lupus erythematosus (LE) is a severe autoimmune disease characterized by the presence of many abnormal immune cells and a large number of autoantibodies and immune complexes, all of which lead to damage to multiple organs, such as the skin, kidney, and brain[1]. Clinically, LE is mainly divided into two types. One type is known as cutaneous lupus erythematosus (CLE), which mainly presents as dermatological injuries and does not involve systemic damage[2]. The other type of LE involves systemic manifestations, including cutaneous, respiratory, renal, cardiovascular and other symptoms, and is called systemic lupus erythematosus (SLE)[3]. Discoid lupus erythematosus (DLE) accounts for >80% of cases of CLE and is the most common type of CLE[4]. On the one hand, approximately 5% of CLE patients will convert to SLE[5]; on the other hand, ~20% of SLE patients exhibit CLE lesions at the time of diagnosis or in the years after diagnosis[6]. Although the

manifestations of cutaneous lesions are distinct between DLE and SLE, the cell composition and the underlying molecular events in cutaneous lesions of DLE and SLE remain unclear.

Recently, single-cell RNA sequencing (scRNA-seq) studies were performed in peripheral blood mononuclear cells (PBMCs), kidney biopsies and lesions of lupus nephritis (LN) patients using a microfluidic device and CEL-Seq2. These results indicated single-cell transcriptional maps of peripheral blood and damaged organs in SLE and identified the important function of type I interferon genes in the pathogenesis of SLE[7–9]. However, due to the throughput limitation of scRNA-seq on microfluidic chips, little information has been obtained about the cutaneous lesions of lupus patients. Besides, previous studies reported that the infiltration of plasmacytoid dendritic cells (pDCs) contributed to the overproduction of type I interferons, which

[1]Department of Dermatology, Hunan Key Laboratory of Medical Epigenomics, Second Xiangya Hospital, Central South University, 410011 Changsha, China. [2]Research Unit of Key Technologies of Diagnosis and Treatment for Immune-related Skin Diseases, Chinese Academy of Medical Sciences, 410011 Changsha, China. [3]Institute of Dermatology, Chinese Academy of Medical Sciences and Peking Union Medical College, 210042 Nanjing, China. [4]Department of Dermatology, Xiangya Hospital, Central South University, 410008 Changsha, China. [5]Department of Pharmacy, The Third Xiangya Hospital, Central South University, 410011 Changsha, China. [6]These authors contributed equally: Meiling Zheng, Zhi Hu, Xiaole Mei, Lianlian Ouyang, Yang Song. ✉e-mail: qianlu5860@pumcderm.cams.cn; zhaoming307@csu.edu.cn

have an important function in the cutaneous lesions of DLE and SLE[10,11]. Several bulk RNA-seq analyses also showed enhanced activation of the interferon (IFN) signaling pathway, innate immune response pathways (including TLR, RIG-I, cytosolic DNA sensing, JAK/STAT) and high expression of some genes, such as *GBP2*, *HLA-F*, *IFIT2*, *RSAD2*, and *ISG15*, in cutaneous lesions of DLE patients[12,13]. In addition, ultraviolet light (UV), immune cells, cytokines and the deposition of immunoglobulins have been reported to have a function in the development of skin inflammation and damage in SLE[14]. Macrophages and *TNF* and *IFN/IFNR* signaling participate in skin injury of SLE[15]. Although previous studies have provided general knowledge about the cutaneous lesions of lupus patients, an understanding of the precise pathological cell types in skin lesions, the functions and activities of each cell subset, and skewed communication networks have not been completely elucidated.

In this study, to further explore the differences between cutaneous lesions of DLE and SLE at a high-throughput single-cell transcriptional level, we collected 23 skin tissues from DLE and SLE patients and age- and sex-matched healthy controls (HCs), separated their epidermis and dermis and performed scRNA-seq on the single-cell suspensions using the Chromium System of 10× Genomics. We show important and detailed information on the differences between healthy skin tissues and skin biopsies of lupus patients and demonstrate the similarities and differences in the cutaneous lesions of DLE and SLE.

## Results

### Sample collection and scRNA-seq workflow

As the cell diameters of epidermal and dermal cells are different, to avoid the capturing preference of the 10× Genomics system, we separated the epidermis and dermis of skin tissues to obtain more transcriptional messages from the cells. In total, 23 skin biopsy samples from 8 DLE patients with an average age of 40.4 ± 11.0 years, 10 SLE patients with an average age of 41.4 ± 12.3 years and 5 HCs with an average age of 32.4 ± 6.0 years were collected. We successfully performed scRNA-seq on 14 epidermal single-cell suspensions (4 HCs, 5 DLE, and 5 SLE samples) and 16 dermal single-cell suspensions (4 HCs, 5 DLE, and 7 SLE) by the 10× Genomics Chromium system (Fig. 1a). The age, sex and other clinical information of all samples are summarized in Supplementary Data 1. The initial number of captured cells per skin biopsy sample, the average reads per cell and other sequencing information are shown in Supplementary Data 2.

### Distinct cell compositions of DLE and SLE illustrated by scRNA-seq

After removing cell doublets, correcting for batch effects, and filtering low-quality cells (see Materials and methods), we acquired 107,428 epidermal cells and 191,701 dermal cells for the downstream analysis (Supplementary Fig. 1a, b, e, f). Through t-stochastic neighborhood embedding (t-SNE) clustering and analysis of differentially expressed genes (DEGs), all epidermal clusters were identified as keratinocytes by high expression of *KRT14* and *KRT1*[16], T cells by *CD3D* and *CD3G*, macrophages/dendritic cells (Macro/DCs) by *LYZ* and *AIF1*[17], melanocytes by *PMEL* and *MLANA*[17], NK cells by *XCL2* and *NKG7*[18] and B cells by *CD79A* and *MS4A1*[19] (Fig. 1b, d, Supplementary Fig. 1d, and Supplementary Data 3). In addition to T cells, Macro/DCs, B cells and NK cells, other dermal clusters were labeled as fibroblasts by *COL1A1* and *COL3A1*[20], endothelial cells by *CDH5* and *VWF*[17], mast cells by *TPSAB1* and *TPSB2*[21] and Schwann cells by *CDH19* and *MPZ*[22] (Fig. 1c, e, Supplementary Fig. 1h, and Supplementary Data 4). Overall, except for immune cells, other cell types identified in our scRNA-seq were consistent with those cell types found in full-thickness skin[23].

The proportions of T cells, B cells, Macro/DC, and NK cells were 29.5%, 1.7%, 10.2%, and 3.5%, respectively, in the epidermis of DLE patients, and 31.2%, 0.4%, 8.6%, and 1.9%, respectively, in the epidermis of SLE patients, which were higher than those in the epidermis of HC

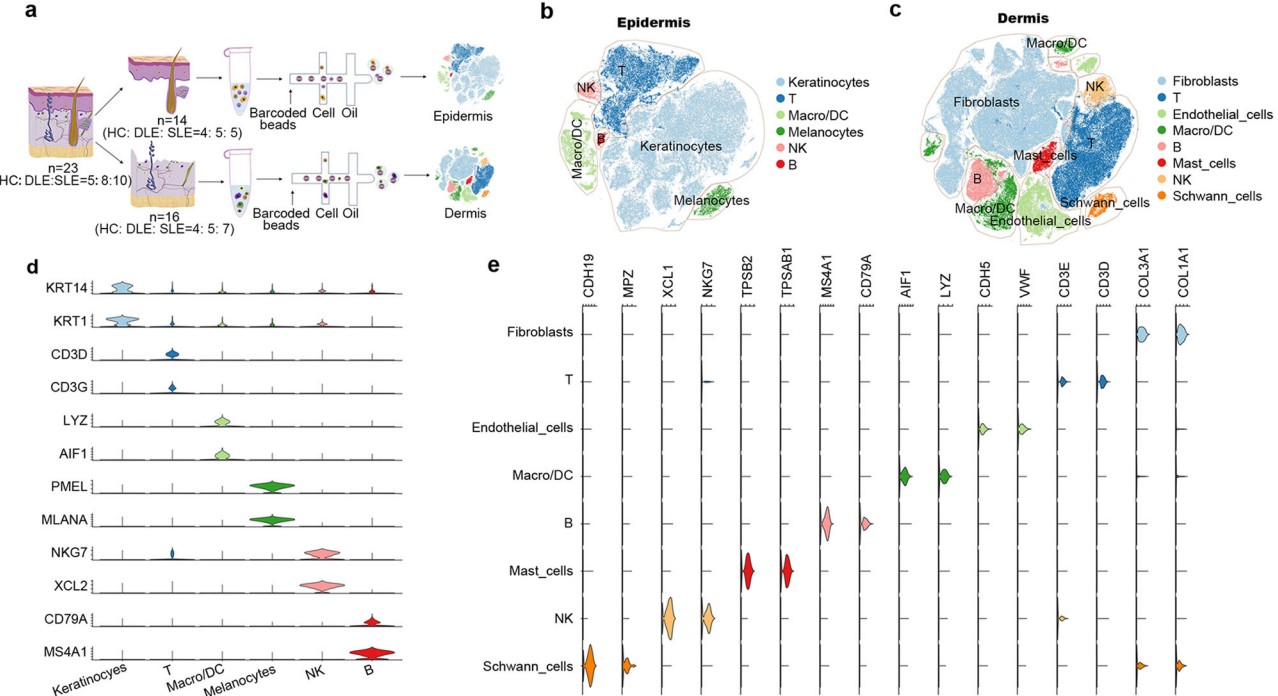

**Fig. 1 | scRNA-seq shows the main cell types in epidermis and dermis.**
**a** Workflow of overall study design. Single-cell RNA sequencing was performed on 14 epidermal samples and 16 dermal samples. **b, c** t-SNE plots represent the main cell types identified in epidermal tissues (**b**) and dermal tissues (**c**). These cell types were identified by the specific gene expression shown in Fig. 1d, e, Supplementary

Data 3 and Supplementary Data 4. Macro/DC: macrophages/dendritic cell.
**d, e** Stacked violin plots show canonical gene expression in each cell type of epidermis (**d**) and dermis (**e**). The violin chart is colored by cell classification. The height represents the level of gene expression, and the width represents the ratio of gene expression. Source data are provided as a Source Data file.

(T: 14.0%, B: 0.1%, Macro/DC: 1.8%, NK: 0.5%) (Fig. 2a). Moreover, the proportions of T cells, B cells and NK cells in the dermis of DLE patients were 43.7%, 16.7%, and 5.4%, respectively, which were higher than those in the dermis of SLE patients (T: 17.2%, B: 2.5%, and NK: 1.3%). The higher proportion of B cells in the lesional skin of DLE patients compared to SLE patients was consistent with a previous report[24]. The proportions of dermal immune cells from both DLE and SLE patients were higher than those from HC (T: 11.5%, B: 0.8%, NK: 0.8%) (Fig. 2b). Regarding dermal Macro/DCs, the proportion in the dermis of SLE patients (8.3%) was higher than that in the dermis of DLE patients (6.7%), and HC (6.9%) (Fig. 2b). In addition, the compositions of cell types in individual samples further supported the findings of different cell proportions in HC, DLE patients and SLE patients (Fig. 2c and Supplementary Fig. 1c, g). Immunofluorescence staining of T cells (CD3), B cells (CD19) and NK cells (CD56) verified the differences in cell composition among HCs, DLE patients and SLE patients (Fig. 2d). Taken together, these results show that immune cells, such as T, B, and NK cells, account for higher cell proportions in DLE patients than in SLE patients or HCs.

## CCL20⁺ keratinocyte and other amplified keratinocyte subtypes in epidermal samples of DLE and SLE

Keratinocytes, the most prominent cell type in the epidermis, have been reported to be involved in the pathogenesis of various autoimmune skin diseases, including lupus[25]. Here, after removing the batch effect (Supplementary Fig. 2a, b), we identified and labeled 12 keratinocyte subtypes: 6 differentiating keratinocyte subtypes (Diff kera-1–6) with high expression of *KRT1*, *KRTDAP*, and *SBSN*[26], 2 basal keratinocyte subtypes (Basal kera-1,2) with high expression of *COL17A1*, *DST*, and *KRT15*[26], 1 channel-related keratinocyte subtype (Channel-related kera) with high expression of components of channel ATPases (*ATP1B1*) and *GJB2*[16], 1 cycling keratinocyte subtype (Cycling kera) with high expression of mitotic markers (*UBE2C* and *TOP2A*[26]) and proliferative markers (*MKI67* and *PCNA*), 1 CCL20⁺ keratinocyte subtype (CCL20⁺ kera) with high expression of CCL20 and 1 terminally differentiating keratinocyte subtype (Terminal diff kera) with high expression of terminal keratinocyte markers (*LCE1A* and *LCE1B*) (Fig. 3c, d). Among all subtypes of keratinocytes, we found that keratinocytes from DLE and SLE groups accounted for more than 75% of Diff kera-3, CCL20⁺ kera, Diff kera-4, Diff kera-5 and Diff kera-6, which suggested their amplifications in the epidermis of DLE and SLE patients (Fig. 3b and Supplementary Fig. 2c).

To identify the transcriptome differences between those DLE and SLE-expanded keratinocyte subtypes, we adopted Gene Ontology (GO) analysis based on the DEGs. Notably, Diff kera-3, characterized by high expression of *S100A6* and *ACTN1*, showed enrichment in epithelial cell proliferation, regulation of actin cytoskeleton organization and muscle organ development (Fig. 3d, f). CCL20⁺ kera highly expressed CCL20 gene (Fig. 3d), which has been reported to be a ligand for the chemokine receptor CCR6 to recruit immune cells[27]. The immunofluorescence staining of CCL20 and KRT10 further confirmed the expansion of CCL20⁺ kera in the epidermis of DLE and SLE patients (Fig. 3e). Furthermore, Diff kera-4 was characterized by high expression of the interferon-stimulated genes (ISGs) *IFIH1*, *IFIT2*, and *IFITM3*, the chemokines *CXCL10* and *CXCL11*, and antigen processing and presentation related molecules *HLA-DRA* and *HLA-DRB1* (Fig. 3d, f and Supplementary Data 5–6). Diff kera-5 with high expression of *CYP1B1* and *GPX4* was involved in the regulation of lipid metabolic processes (Fig. 3f and Supplementary Data 5–6). Diff kera-6 expressed the chemokine ligands *CCL4* and *CCL5*, and the chemokine receptor *CXCR4*, which enriched leukocyte cell adhesion and migration (Fig. 3d, f and Supplementary Data 5–6). Interestingly, the pathways of neutrophil activation and neutrophil degranulation, which have been reported to have an important function in the development of lupus[28], were commonly enriched in CCL20⁺ kera, Diff kera-4, Diff kera-5 and Diff kera-6 (Fig. 3f and Supplementary Data 5–6).

Due to the existence of similar and distinct GO enrichments in the lupus-expanded keratinocyte subtypes, we wondered whether there was a differentiating association between those subtypes. Thus, we performed pseudotime trajectory analysis by Monocle2. Intriguingly, the results showed a potential differentiated trajectory from Diff kera-2 to Diff kera-1, Diff kera-6 and finally to Terminal diff kera or CCL20⁺ kera and Diff kera-4 (Fig. 3g). Diff kera-2 was located in the initial position of the trajectory with high expression of genes such as *GRHL3* and *SERPINB2*. Both Diff kera-1 with high expression of *ID1*, *FGFR3*, and *CHP2* and Diff kera-6 with high expression of *RUNX1*, *RGS1*, and *CXCR4* was located in the middle of the trajectory. Terminal diff kera was in one end stage of the trajectory with high expression of *LCE1C*, *LCE2B*, and *LCE1B*. CCL20⁺ kera and Diff kera-4 were located in the other end stage with high expression of *IFI3*, *IFITM1*, and *CXCL10* (Fig. 3h). The above findings suggested a potential differentiation direction of keratinocytes in lesional skin of lupus.

## CXCL1⁺ fibroblasts, HLA⁺ fibroblasts and other expanded fibroblast subtypes in the dermis of SLE patients

Recent reports have shown that fibroblasts participate in skin wound healing and the progression of scleroderma[29,30]. However, the function of fibroblasts in lupus is poorly understood. In our study, after integrating 92,162 fibroblasts from DLE patients, SLE patients and HCs and removing batch effects, we generated and labeled 9 fibroblast subtypes (Fig. 4a and Supplementary Fig. 3a–c). The cells from SLE occupied for more than 75% of CXCL1⁺ Fib, HLA⁺ Fib1, Pericyte2, and Fib4, which indicated their amplification in the dermis of SLE (Fig. 4b and Supplementary Fig. 3d). Although we were unable to identify fibroblasts by the known marker genes of the classic fibroblast subtype[31,32] (Supplementary Fig. 3e), two fibroblast genes, *COL1A1* and *DCN*[20], were highly expressed in 7 fibroblast subtypes (Fib1, Fib2, CXCL1⁺ Fib, HLA⁺ Fib1, Fib3, Fib4, and HLA⁺ Fib2). Pericyte1 and Pericyte2 were characterized by high expression of ACTA2 and TPM2, which indicated a pericyte subtype (a special subtype of fibroblasts[32]) (Fig. 4c, d). CXCL1⁺ Fib was found to have high expression of *CXCL1*, which has been reported to promote the infiltration of immune cells, especially neutrophils[33] (Fig. 4c, d). In addition, we identified a marker gene *CXCL12* of activated fibroblasts[29] to be highly expressed in CXCL1⁺ Fib and HLA⁺ Fib2. Interestingly, major histocompatibility complex-II family member genes (*HLA-DRB1* and *HLA-DRA1*), which are usually expressed in antigen-presenting cells, were identified to be highly expressed in HLA⁺ Fib1 and HLA⁺ Fib2, suggesting that these fibroblasts may function as nonclassical antigen-presenting cells (APCs) (Fig. 4c, d and Supplementary Data 7). Although the CXCL1⁺ Fib and HLA⁺ Fib1 subtypes were not expanded in DLE, we observed that *CXCL1* and *HLA-DRB1* were highly expressed in DLE fibroblasts (Supplementary Fig. 3f). The immunofluorescence staining of *CXCL1* and *HLA-DRB1* with Vimentin also verified their overexpression in SLE and DLE (Fig. 4e, f).

Furthermore, GO function analysis based on DEGs in expanded fibroblast subtypes showed that the biological processes (BPs) of response to oxygen levels, response to hypoxia and neutrophil degranulation were enriched in HLA⁺ Fib1, Fib4, and CXCL1⁺ Fib cells, while the regulation of viral process and the immune response-activating signal transduction were enriched in HLA⁺ Fib1 and Fib4 cells. In addition, HLA⁺ Fib1 and CXCL1⁺ Fib showed enrichment of leukocyte cell-cell adhesion, positive regulation of leukocyte activation and positive regulation of cytokine production. Strikingly, type I interferon signaling pathway and response to virus were separately enriched in Pericyte2 and CXCL1⁺ Fib (Fig. 4g and Supplementary Data 8).

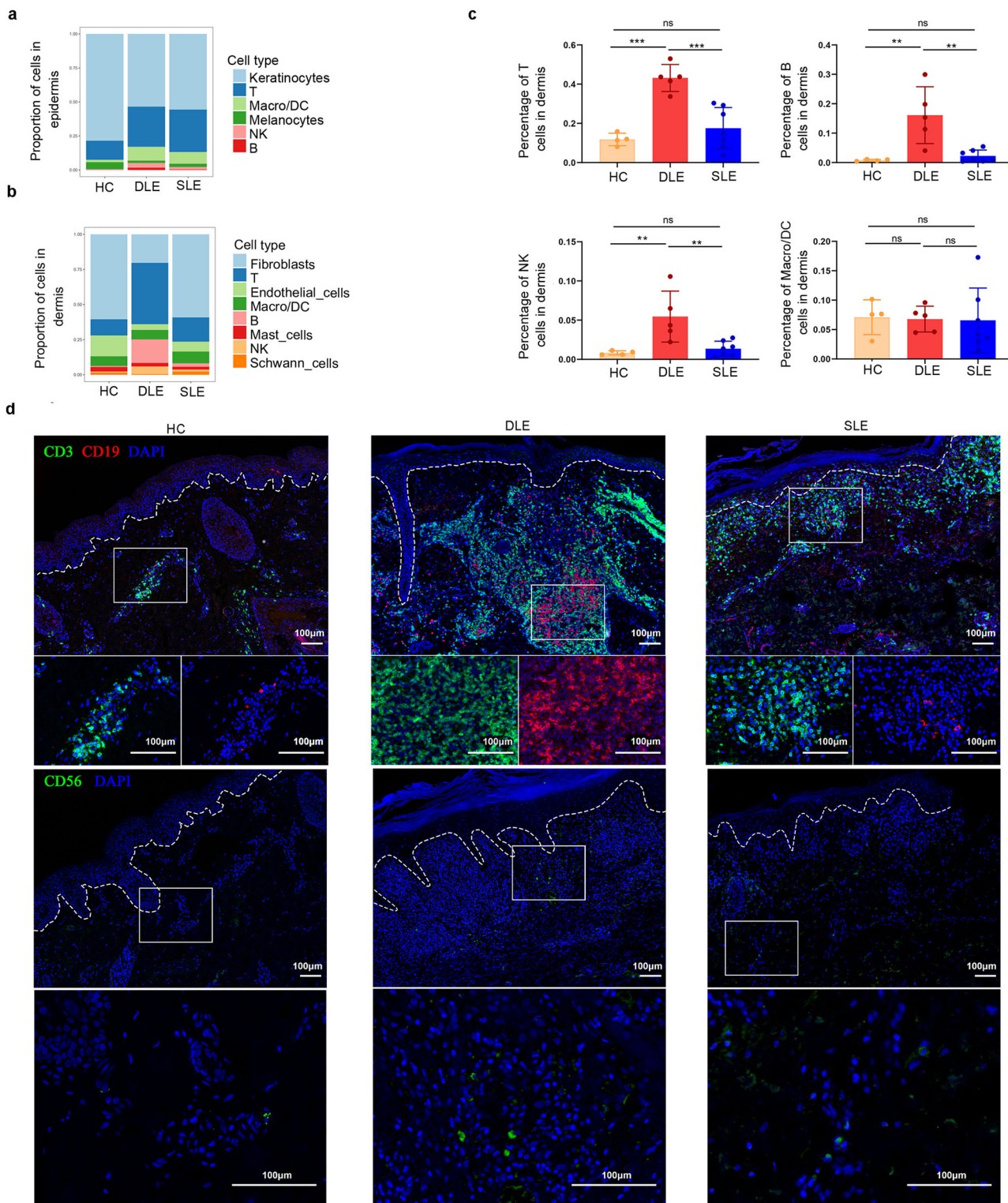

**Fig. 2 | Distinct cell compositions of HC, DLE, and SLE. a, b** Bar plots show the proportions of 6 cell types in the epidermis (**a**) and 8 cell types in the dermis (**b**) of biopsy samples from DLE and SLE patients. Macro/DC: macrophages/dendritic cell. **c** Histogram plots comparing the proportions of the main immune cell types, including T cells, B cells, Macro/DCs and NK cells in dermis of HCs ($n = 4$), DLE patients ($n = 5$), and SLE patients ($n = 7$). The histograms for HCs are shown in pink, histograms for DLE are in red and histograms for SLE are in blue. Data were presented as mean ± SD, the P values were calculated using a one-way ANOVA test with Bonferroni's multiple comparisons test. *$P < 0.05$, **$P < 0.01$, ***$P < 0.001$, ns: no

significance. T cells: DLE vs HC, $P = 0.0002$; SLE vs DLE, $P = 0.0004$, B cells: DLE vs HC, $P = 0.0036$; SLE vs DLE, $P = 0.0027$. NK cells: DLE vs HC, $P = 0.0099$; SLE vs DLE, $P = 0.0090$. **d** Represent images show immunofluorescence staining for CD3 (green)/CD19 (red) for T cells/B cells, CD56 (green) for NK cells in HCs, DLE and SLE samples. The dotted line outlines the dermal-epidermal junction. Parts of the stained area were enlarged in the corresponding single-unit grid. Scale bar: 100 μM. Data are representative of three independent experiments. Source data are provided as a Source Data file.

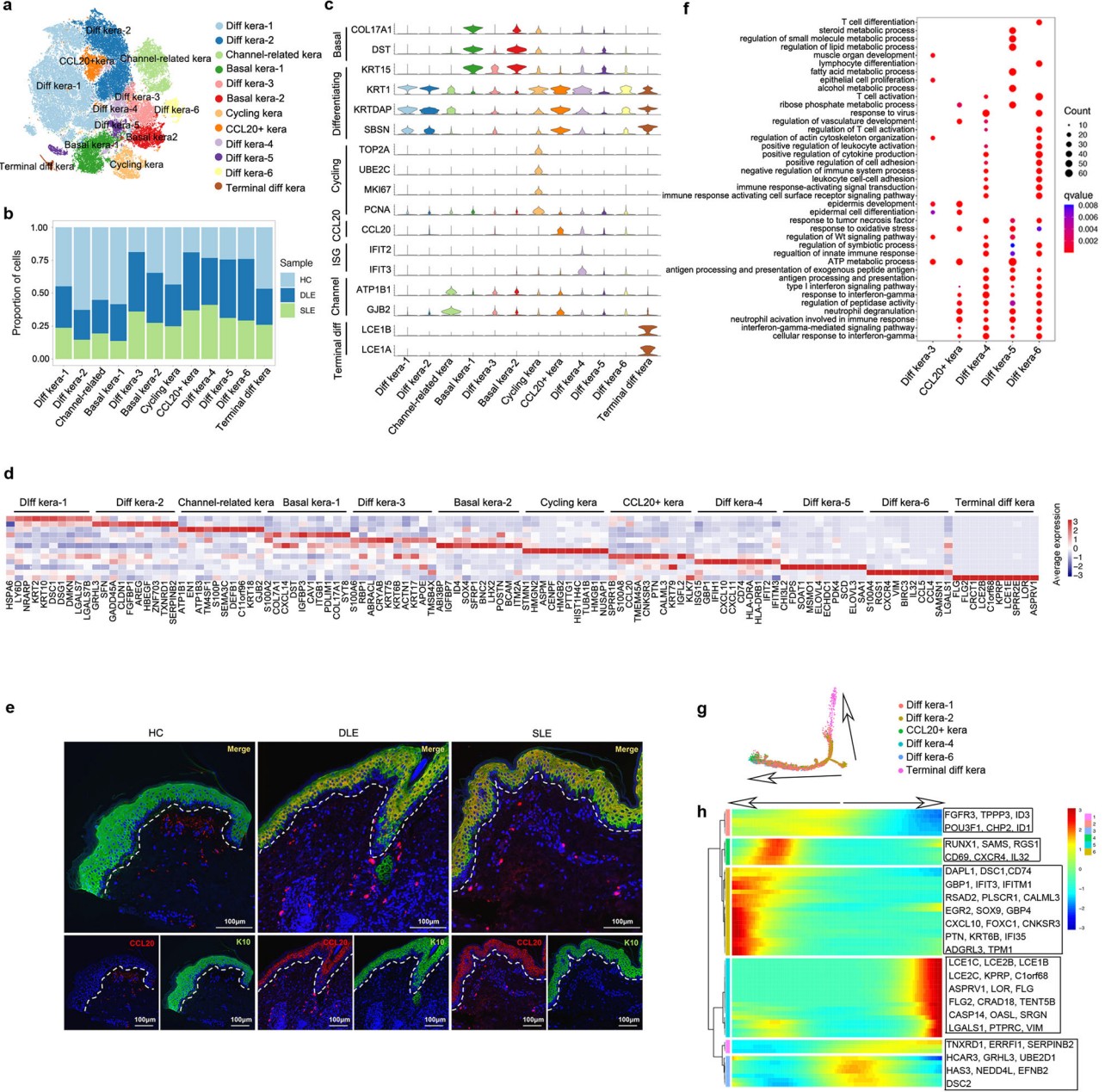

**Fig. 3 | Sub-clustering analysis of keratinocytes identifies major keratinocyte subtypes in epidermis. a** t-SNE plot represents subtypes of keratinocytes merged from HC, DLE and SLE by Seurat packages. Diff kera-1–6: differentiating keratinocyte subtype 1–6; Basal kera-1, 2: basal keratinocyte subtype 1, 2; Channel-related kera: channel-related keratinocyte; Cycling kera: cycling keratinocyte; CCL20+ kera: CCL20+ keratinocyte subtype; Terminal diff kera: terminally differentiating keratinocyte subtype. **b** A bar plot shows the cell proportions of each sample group in labeled keratinocyte subtypes. **c** Stacked violin plots represents the expression levels of the selected genes used for defining keratinocyte subtypes. **d** A heatmap displays the top 10 differentially expressed genes in each subtype of keratinocytes. **e** Immunofluorescence staining represents the expression of CCL20 (red) and

KRT10 (green) in the epidermis of HC, DLE, and SLE. KRT10 staining represents keratinocytes. Scale bar: 100 μM. Data are representative of 3 independent experiments. **f** A bubble diagram shows the Gene Ontology (GO) biological process (BP) terms enriched in expanded keratinocyte subtypes. **g** Trajectory plot shows pseudotime trajectory analysis of selected keratinocyte subtypes by Monocle2. Different colors represent keratinocyte subtypes. The arrow indicates a possible differentiation direction. **h** A heatmap shows gene clusters with differentiation states that were predicted by pseudotime trajectory analysis. The color represents the levels of given gene expression. The arrow indicates a possible differential direction. Source data are provided as a Source Data file.

We also performed pseudotime trajectory analysis on fibroblast subtypes to explore whether there was a differential trajectory in these expanded fibroblast subtypes. The pseudotime trajectory analysis presented a potentially differentiated trajectory of fibroblasts, in which Fib2 was located in the initial part and HLA+ Fib1 and HLA+ Fib2 expressing *HLA-DRB1*, *CD74*, and *HLA-DRA* were located in the first middle stage. Part of the Fib1 expressing *PDGFRA* and *FBLN* was in one

end stage of the trajectory, and CXCL1+ Fib with *CYP1B1* and *CXCL1* expression was located in the other end stage (Fig. 4h, i).

## ISGs^hi T cells, GPR183+ T cells, and other expanded T cell subclusters in cutaneous lesions of DLE and SLE patients

Abnormal T cells in the peripheral blood of patients with lupus have been widely reported[34,35]. However, the T cell subset changes in

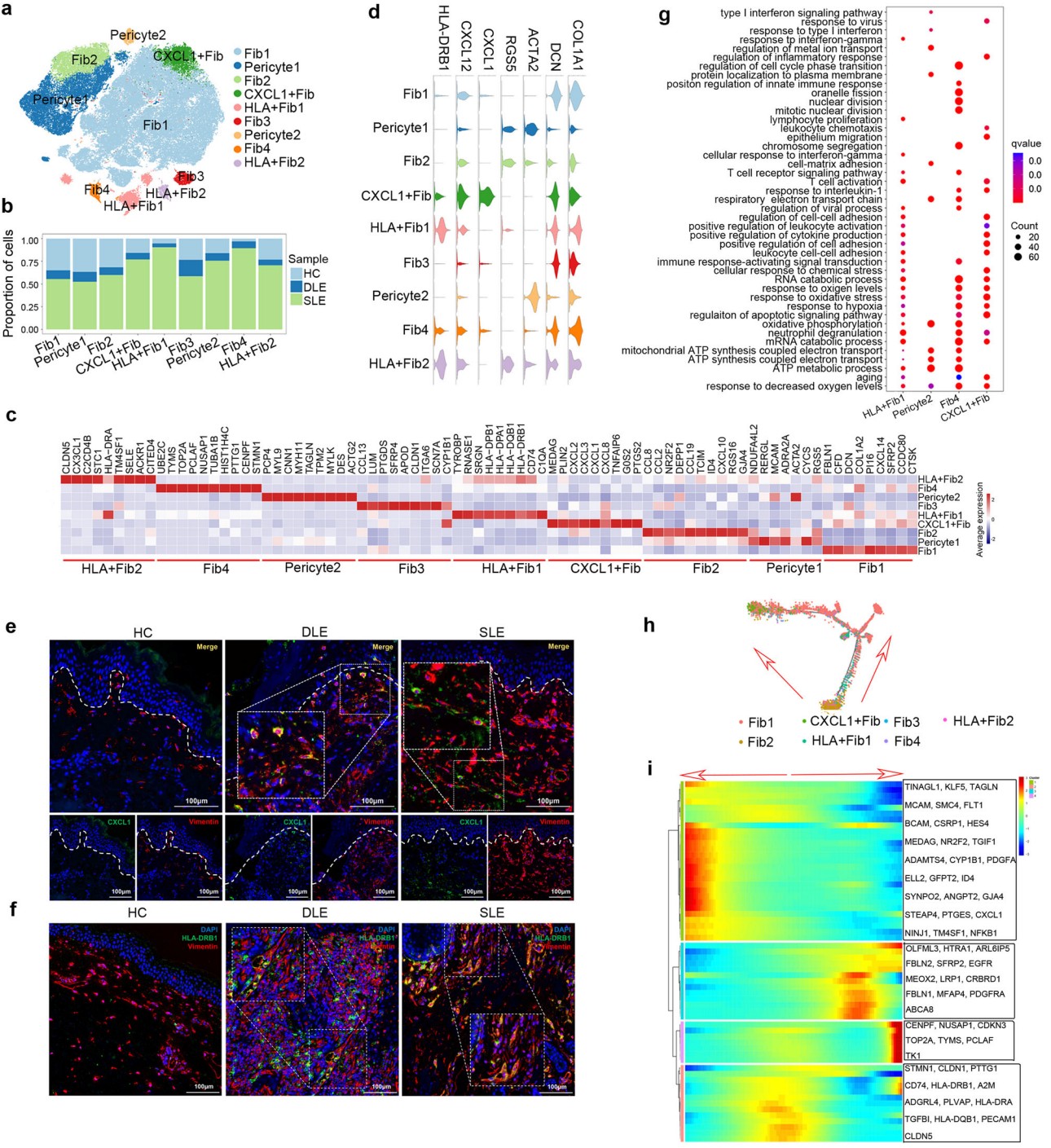

**Fig. 4 | Sub-clustering analysis of fibroblasts identifies major fibroblast subtypes in dermis. a** t-SNE plot shows labeled fibroblasts which were merged from HC, DLE and SLE by Seurat packages. Fib1-4: Fibroblast subtype 1–4; HLA⁺ Fib: HLA⁺ Fibroblast; CXCL1⁺ Fib: CXCL1⁺ Fibroblast. **b** A bar plot shows the sample compositions of fibroblasts subtypes. **c** A stacked violin plot shows selected gene expression used for identifying fibroblast subtypes. **d** A heatmap displays the top 10 differentially expressed genes in each subtype of fibroblasts. **e**, **f** Immunofluorescence staining represents the expression of CXCL1 (green) (**e**), HLA-DRB1(green) (**f**) and Vimentin (red) in the dermis of HC, DLE, and SLE. The immunofluorescence staining of Vimentin represents the cell type of fibroblasts. Scale bar: 100 μM. Data are representative of 3 independent experiments. **g** A bubble diagram shows the Gene Ontology (GO) biological process (BP) terms of expanded fibroblast subtypes. **h** Trajectory plot shows pseudotime trajectory analysis of selected fibroblast subtypes. The arrow indicates the possible differentiation direction. **i** Heatmaps show gene clusters with the differentiation state that were predicted by pseudotime trajectory analysis. The arrow indicates the possible differentiation direction. Source data are provided as a Source Data file.

cutaneous lesions of DLE and SLE are not fully understood. To evaluate T cell subset transcriptome changes, we integrated a total of 23,982 epidermal T cells (HC: 4,286; DLE: 11,429; SLE: 8,267) and yielded 7 T cell subclusters (SCs) (T_SC0 to T_SC6) (Fig. 5a and Supplementary Fig. 4a–c). The cells from DLE and SLE accounted for more than 75% of the 5 epidermal T cell SCs (T_SC1-T_SC5) (Fig. 5b and Supplementary Fig. 4d). In addition, a total of 40,047 dermal T cells (HC: 5,330; DLE: 20,327; SLE: 14,390) yielded 7 dermal T cell SCs (T_SC0-T_SC6) (Fig. 5f and Supplementary Fig. 4f–h). DLE and SLE accounted for 75% of all dermal T cell SCs except T_SC2 (Fig. 5g and Supplementary Fig. 4i).

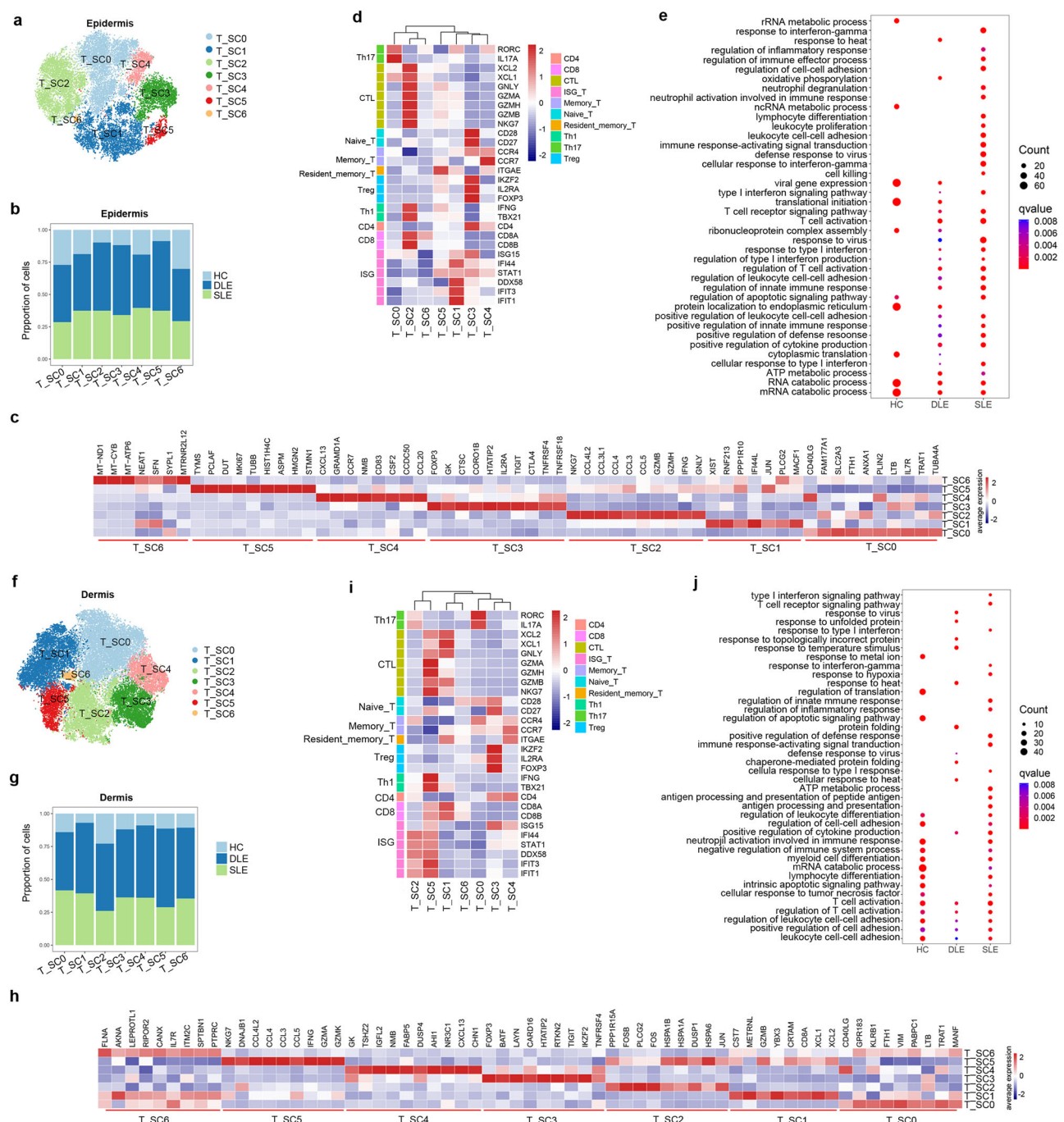

**Fig. 5 | Sub-clustering analysis of T cells identifies major T cell subclusters in skin tissues. a, f** t-SNE plots distribution of T cell subclusters (SCs) in epidermis (**a**) and in dermis (**f**). **b, g** Bar plots show the proportions of cells from HC, DLE, and SLE in T cell SCs of epidermis (**b**) and dermis (**g**). **d, i** Heatmaps show the specific classic gene expression in SCs of epidermal (**d**) and dermal (**i**) T cells. The annotation of rows shows the classical subtypes of T cells with marker genes. CTL: Cytotoxic T Lymphocyte; ISG: interferon-stimulated gene. **c, h** Heatmaps indicating the top 10 differentially expressed genes between each SC of T cells in the epidermis (**c**) or dermis (**h**). **e, j** Bubble diagrams show the enriched Gene Ontology (GO) biological process (BP) terms in HC, DLE and SLE group of epidermis (**e**) and dermis (**j**). Source data are provided as a Source Data file.

According to the classical marker genes and DEGs profile, there were several T cell SCs with similar characteristics in both epidermis and dermis. First, ISGs (*IFI6*, *IFI44*, *IFIH1*, and *DDX58*) were highly expressed in epidermal T_SC1 and dermal T_SC5, which are identical to the previously reported T-SC4 in PBMCs of lupus patients[9] (Fig. 5c, d, h, i, Supplementary Fig. 4e, j, and Supplementary Data 9). Moreover, a gene panel of ISGs (*IFI6*, *IFI44*, *IFITM3*, *ISG20*, *IFI27*, *ISG15*, and *IFI44L*) showed significantly higher expression in dermal T cells of DLE and SLE patients than in HCs (Supplementary Fig. 4k). Second, epidermal

T_SC3 and dermal T_SC3 were both identified by high expression of Treg marker genes (*IKZF2*, *IL2RA* and *FOXP3*[36]; Fig. 5c, d, h, i and Supplementary Data 9). Epidermal T_SC2 and dermal T_SC5 highly expressed marker genes of Th1 (*IFNG* and *TBX21*[37]) and CTLs (cytotoxic T cells: *XCL2*, *XCL1*, and *GZMA*[9]) (Fig. 5d, i), which suggests the potential effector functions of these two T cell subsets. Notably, *GPR183*, which has been reported to be involved in the migration, expansion, and infiltration of B cells and T follicular helper cells (Tfh)[38,39], showed high expression in dermal T_SC0 (Fig. 5h), suggesting that this T cell subset

may be related to B cell accumulation and ectopic germinal center formation in cutaneous lesions of DLE and SLE.

To determine the distinct transcriptome functions of T cells in skin tissues among HCs, DLE patients and SLE patients, we conducted GO analysis on DEGs. Intriguingly, T cells in the epidermis of DLE and SLE patients showed enrichment of some BPs, including cellular response to type I interferon, positive regulation of defense response, positive regulation of leukocyte cell-cell adhesion and type I interferon signaling pathway, which were distinct from T cells in the epidermis of HC with enrichment of RNA catabolic process and translational initiation (Fig. 5e). In contrast to epidermal T cells, both cell adhesion and regulation of T cell activation were enriched in dermal T cells of DLE patients, SLE patients and HCs. However, response to virus and cellular response to heat were specifically enriched in dermal T cells from DLE, and type I interferon signaling pathway, response to interferon gamma and regulation of inflammatory response were enriched in dermal T cells from SLE (Fig. 5j). These GO BP results suggest the common and distinct functions of epidermal and dermal T cells among HCs, DLE patients and SLE patients.

To further understand the relationship among dermal T cells of HCs, DLE patients and SLE patients, we performed pseudotime trajectory analysis to assess the potential differentiation of dermal T cells. The trajectory results showed that dermal T cells of HC were located in the initial stage with high expression of *GAS5*, *TLE5* and *SNHG29*, dermal T cells highly expressing ISGs (*IFITM1*, *IFI44L*, *ISG15*, and *IFITM3*) from DLE and SLE were located in the middle stage, and dermal T cells mainly from SLE with high expression of *H1FX*, *HLA-A*, and *TGFBR2* were located in the end stage (Supplementary Fig. 4l, m).

### ISG^hi plasma, HSP^hi B and other expanded B cell SCs in cutaneous lesions of DLE and SLE patients

B cell selection defects and autoreactive B cell activation lead to the overproduction of autoreactive antibodies in lupus[40]. In our study, 864 epidermal B cells (HC: 41; DLE: 678; SLE: 105) yielded 5 B cell SCs (B_SC0-B_SC4), almost all of which were composed of cells from DLE and SLE (Fig. 6a, b and Supplementary Fig. 5a–d). In addition, 9,494 dermal B cells (HC:338; DLE: 7,190; SLE: 1966) generated 7 B cell SCs (B_SC0-B_SC6) (Fig. 6f and Supplementary Fig. 5g–i). Dermal B cells from DLE and SLE accounted for more than 75% in each dermal B cell SC (Fig. 6g and Supplementary Fig. 5j), which is consistent with a previous report of B cell expansion in CLE lesions[24,41].

To clarify the characteristics of epidermal and dermal B cell subclusters, we represented the classic B cell subset markers and DEGs expression in B cell SCs. Notably, B cell SCs from the epidermis and dermis presented similar features. We found high expression of ISGs in epidermal B_SC3 and dermal B_SC4 and B_SC5, which were identified as plasma cells by the gene expression of *CD38*, *IGHG4*, and *IGHG1*[42] (Fig. 6c, d, h, i and Supplementary Data 10). These three SCs were similar to the P-SC0 in the PBMCs of lupus patients identified in a previous scRNA-seq study[9] (Supplementary Fig. 5e, k). Hsp70-coding genes, which are involved in antigen presentation[43,44], were highly expressed in epidermal B_SC0, dermal B_SC3 and B_SC6 (Fig. 6d, h), suggesting that these B_SCs may participate in antigen presentation. Age-associated B cells (ABC) have been reported to be abnormally expanded in SLE[7,45]. Therefore, we analyzed the expression of ABC marker genes (*ITGAX*, *TBX21*, *FCRL2*[46]) in B cell subsets and found that these marker genes were expressed in multiple B cell subsets but not in a specific subset (Supplementary Fig. 5f, l).

Next, we performed GO analyses on epidermal and dermal B cells of HCs, DLE patients and SLE patients. The results showed that epidermal B cells in HC were involved in RNA catabolic processes and T cell activation, while epidermal B cells in DLE and SLE were involved in more BPs, including the type I interferon signaling pathway, response to virus, response to interferon gamma, regulation of innate immune response, neutrophil activation involved in immune response and

leukocyte cell-cell adhesion (Fig. 6e). In addition, epidermal B cells of DLE were characterized by macroautophagy, which has been reported previously to be activated in B cells of SLE lesions and required for plasmablast development[47] (Fig. 6e). Different from epidermal B cells, the response to interferon gamma and viral life cycle were enriched in dermal B cells in each sample group. Moreover, dermal B cells from HCs and DLE patients were enriched in lymphocyte differentiation, leukocyte cell-cell adhesion and T cell activation, and only dermal B cells from SLE patients were enriched in the type I interferon signaling pathway (Fig. 6j). Taken together, these findings indicate that epidermal and dermal B cells, including ISG^hi plasma cells and HSP^hi B cells, were expanded in the epidermis and dermis of DLE and SLE, especially in the dermis of DLE.

### pDC, ISGs^hi Macro/DC and other expanded Macro/DC SCs in cutaneous lesions of DLE and SLE patients

Macrophages and DCs have been reported to be defective in clearing apoptotic cells in lupus[48]. In our study, 6,595 epidermal Macro/DCs (HC: 546; DLE: 3832; SLE: 2217) and 11,864 dermal Macro/DCs (HC: 3059; DLE: 2874; SLE: 5931) were used for sub-clustering analysis (Supplementary Fig. 6a, e). More than 75% of the 5 epidermal Macro/DC SCs, except M_SC5 and dermal M_SC3, M_SC5 and M_SC6, were contributed by the DLE and SLE samples (Fig. 7b, g, Supplementary Fig. 6b–d, f–h).

To discover transcriptional differences among Macro/DC SCs of HCs, DLE patients and SLE patients, we performed DEGs analysis. We found that ISGs were highly expressed in epidermal M_SC1 cells, which were identified as a mix of macrophages and DCs because of the co-expression marker genes of CD16^+ DCs (*FCGR3A*, *CXCL10*, *CXCL11*[41]) and macrophages (*MRC1*, *CD163*, *CD68*[49]) (Fig. 7d). Dermal M_SC6 highly expressed ISGs and marker genes of CD16^+ DCs (Fig. 7i). Epidermal M_SC2 and dermal M_SC3 showed high expression of marker genes of pDCs (*JCHAIN* and *MZB1*[48]) (Fig. 7c, d, h, i, Supplementary Data 11), which have been reported to accumulate and help produce spontaneous germinal centers in lupus[50]. As classic APCs, multiple Macro/DC SCs expressed HSP70-coding genes. In addition, epidermal-specific Langerhans cells were identified by *CD207* expression in epidermal M_SC5 (Fig. 7d).

Further GO analysis showed that antigen processing and presentation, neutrophil activation involved in immune response and T cell activation were common enriched in epidermal Macro/DCs in HCs, DLE patients and SLE patients. The differences were that epidermal Macro/DCs of DLE contained some BPs, including cell killing, positive regulation of leukocyte cell-cell adhesion and response to oxidative stress, and epidermal Macro/DCs of SLE were enriched in the type I interferon signaling pathway (Fig. 7e). Dermal Macro/DCs from HCs, DLE patients and SLE patients were all involved in antigen processing and presentation and T cell activation, which was similar to epidermal Macro/DCs. Although type I interferon signaling pathway, type I interferon production and response to virus were enriched in dermal Macro/DCs of both DLE and SLE, macroautophagy, which was reported to be impaired in SLE macrophages[51], and regulation of inflammatory response were enriched in dermal Macro/DCs of SLE (Fig. 7j). These differences in Macro/DCs further suggest a possible difference in the pathogenesis of cutaneous lesions between DLE and SLE.

### ISGs and HSP-coding gene overexpression in NK cells of DLE and SLE patients

NK cells have rarely been reported in cutaneous lesions of lupus. In our study, 2152 epidermal NK cells (HC: 214; DLE: 1398; SLE: 540) and 3489 dermal NK cells (HC: 282; DLE: 2165; SLE: 1042) were ultimately involved in sub-clustering analysis (Supplementary Figure 7a, e). Except for epidermal and dermal N_SC3, cells from DLE and SLE composed more than 75% of other epidermal and dermal NK SCs (Fig. 8b, g and Supplementary Fig. 7b–d, f–h).

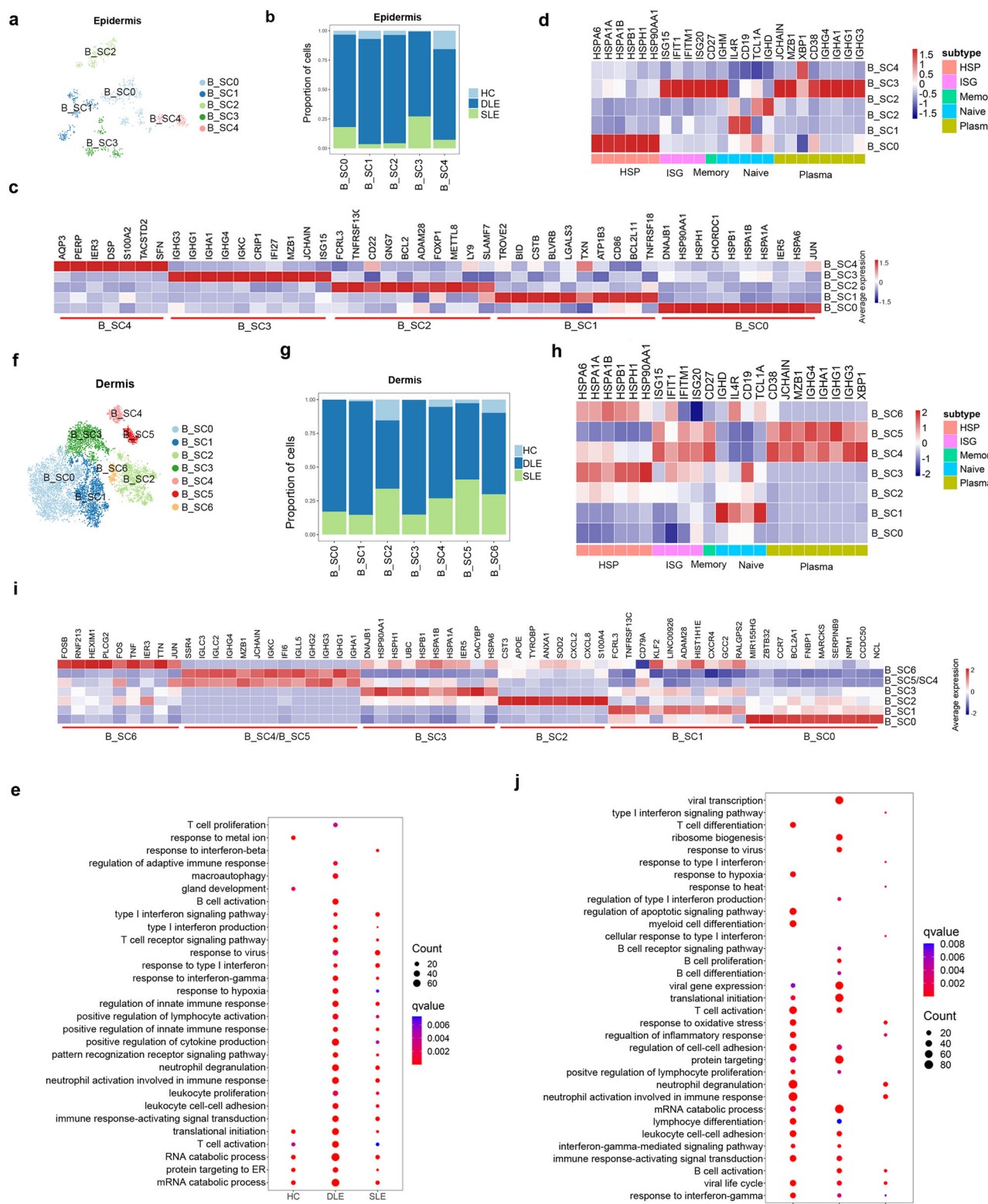

**Fig. 6 | Sub-clustering analysis of B cells implicates major B cell subclusters.** **a**, **f** t-SNEs plot B cell subclusters (SCs) distribution in epidermis (**a**) and dermis (**f**). **b**, **g** Bar plots show the proportions of cells from HC, DLE and SLE in B cells SCs of epidermis (**b**) and dermis (**g**). **c**, **h** Heatmaps show the top 10 differentially expressed genes in 5 epidermal B cell SCs and 7 dermal B cell SCs. **d**, **h** Heatmaps show specific classical genes expression in B cell SCs of epidermis (**d**) and dermis

(**h**). The annotation of rows shows the classical subtypes of B cells with marker genes. HSP: Heat Shock Protein coding gene. ISG: interferon-stimulated gene. **e**, **j** Bubble diagrams show the Gene Ontology (GO) biological process (BP) terms enriched in epidermal or dermal B cells of HC, DLE, and SLE. Source data are provided as a Source Data file.

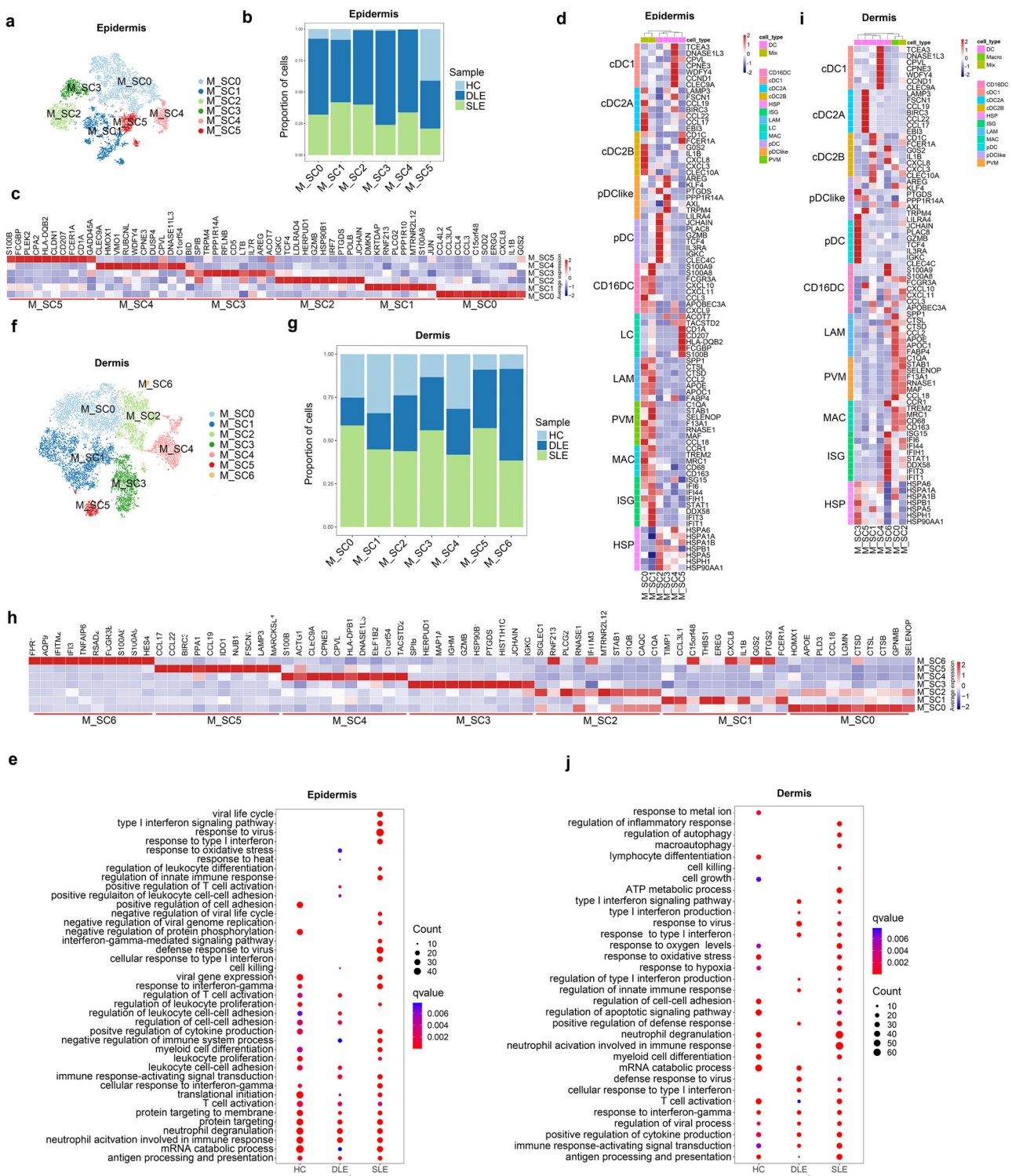

**Fig. 7 | Sub-clustering analysis of macrophage/dendritic cells (Macro/DCs) identifies major Macro/DCs subclusters. a, f** t-SNEs plot epidermal (**a**) or dermal (**f**) Macro/DCs subclusters (SCs) distribution. **b, g** Bar plots show cells proportions of HC, DLE and SLE in six epidermal Macro/DCs SCs (**b**) and in 7 dermal Macro/DCs SCs (**g**). **c, h** Heatmaps display the top 10 differentially expressed genes in 6 epidermal Macro/DCs SCs (**c**) and 7 dermal Macro/DCs SCs (**h**). **d, i** Heatmaps show the expression of specific classic genes in epidermal (**d**) and dermal Macro/DCs SCs (**i**). LC Langerhans, LAM lipid-associated macrophages, PVM perivascular macrophages, MAC macrophages. The annotations of rows indicate the classical subtypes of Macro/DCs with marker genes. **e, j** Bubble diagrams show the Gene Ontology (GO) biological process (BP) terms enriched in epidermal (**e**) and dermal Macro/DCs of HC, DLE and SLE (**j**). Source data are provided as a Source Data file.

According to DEGs analysis, we found that ISGs were more highly expressed in NK cells from DLE and SLE patients than in those from HCs, and HSP70-coding genes were highly expressed in epidermal N_SC0 and dermal N_SC1 cells (Fig. 8c, d, h, i and Supplementary Data12). Further GO functional analysis showed the enrichment of the type I interferon signaling pathway in epidermal NK cells from DLE and SLE patients (Fig. 8e). In contrast to dermal NK cells of SLE patients enriching the type I interferon signaling pathway and the regulation of

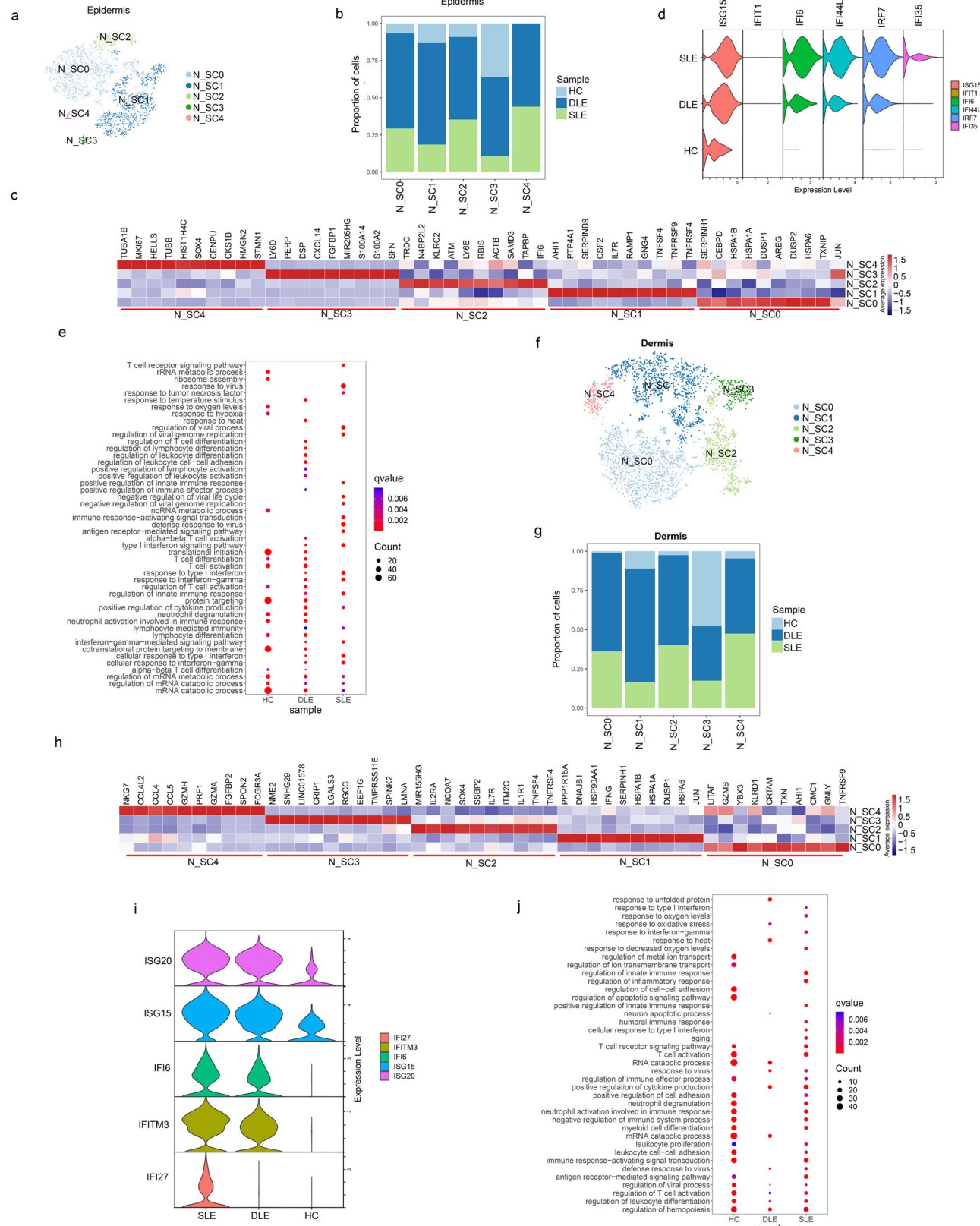

**Fig. 8 | Sub-clustering analysis of NK cells identifies major NK subclusters.**
**a**, **f** t-SNEs plot show cell distribution of epidermal (**a**) and dermal NK subclusters (SCs) (**f**). **b**, **g** Bar plot shows the sample proportions in NK cell SCs of epidermis (**b**) and dermis (**g**). **c**, **h** Heatmaps indicate the top 10 differentially expressed genes in 5 epidermal NK cell SCs (**c**) and 5 dermal NK cell SCs (**h**). **d**, **i** Stacked violin plot show the ISGs expression in sample group of epidermis (**d**) or dermis (**i**). **e**, **j** Bubble diagrams show the Gene Ontology (GO) biological process (BP) terms enriched in epidermal (**e**) or dermal (**j**) NK cells of HC, DLE, and SLE. Source data are provided as a Source Data file.

inflammatory pathway, only the defense response to virus and the positive regulation of cytokine production were enriched in dermal NK cells of DLE patients (Fig. 8j).

### Active cell communication in cutaneous lesions of DLE and SLE patients

The interaction between cells based on receptors and ligands has an important function in physiological and pathological processes[26]. To investigate the interaction among different cell types identified by scRNA-seq in lupus cutaneous lesions, we conducted cell communication analysis by CellPhoneDB[52]. Based on the number of ligand–receptor interaction pairs, we calculated the interaction scores. The cell communication scores of keratinocytes and macro/DCs were significantly higher in the epidermis of DLE and SLE patients than in HCs (Fig. 9a). Notably, the scores of keratinocytes, Macro/DCs and melanocytes in DLE were higher than those in SLE (Fig. 9a). Moreover, the cell communication scores of endothelial cells, fibroblasts, and Macro/DCs in the dermis of DLE were higher than those in SLE and HC (Fig. 9b). These findings suggested a potential connection of keratinocytes, endothelial cells, fibroblasts, and Macro/DCs in the immune microenvironment in cutaneous lesions of DLE and SLE.

Furthermore, the more highly expressed chemokine ligands and receptors of cell types in DLE and SLE are shown in Supplementary Data13. Immune cells, such as T cells, NK cells, Macro/DCs and B cells in the epidermis of DLE and SLE, highly expressed *CXCR3* (Fig. 9c), which is a receptor for the chemokine ligand *CCL20* with higher expression levels in keratinocytes of DLE and SLE patients than in those of HCs. In addition, other ligand-receptor pairs in keratinocytes and immune cells, such as *TNFSF9_TNFRSF9* and SPN_*ICAM1*, were also highly expressed in the epidermis of DLE and SLE (Fig. 9c). Macro/DCs, as antigen-presenting cells, interact most strongly with other immune cells, in which *FAM3C_CLEC2D* indicates the possible interaction between Macro/DCs and T or B cells in DLE and SLE (Fig. 9c). In addition, *FAM_3C_ HLA_C* was expressed at higher levels in immune cells (B, NK, T, Macro/DCs) and keratinocytes of DLE than SLE (Fig. 9c).

Regarding the dermis, the receptors of *CCL19* (*CCR7*, *CCR2*, and *CXCR3*), *CXCL12* (*CXCR4*), *FAM3C* (*CLEX2D*), and *TNFSF13B* (*CD40*) in fibroblasts showed the highest expression in B, Macro/DCs, NK, and T cells in the DLE group among the three sample groups (Fig. 9d). The interaction between Macro/DCs and other cells was represented by the expression of *TNARSF13B_CD40*, which also showed the highest expression in DLE (Fig. 9d). Immunochemical staining verified the increased *CCL19* expression in fibroblasts (vimentin) and the increased *CCR7* expression in macrophages (CD68) in cutaneous lesions of DLE and SLE patients (Fig. 9e). Together, the overexpression of these ligands and receptors may be involved in the activation, migration, and residence of immune cells in cutaneous lesions of DLE and SLE patients.

## Discussion

Lupus is a spectrum of autoimmune diseases, including DLE on one end and SLE on the other[53]. Although we know that there are distinct skin phenotypes between DLE and SLE and understand the basic immune changes that occur in cutaneous lesions, the precise cell composition and their functions in cutaneous lesions in DLE and SLE remain unclear. Here, we identified detailed cell types and their constitutions in the epidermis and dermis of cutaneous lesions in DLE and SLE by scRNA-seq. Our dataset comprised transcriptional database of 107,428 epidermal cells and 191,701 dermal cells from 23 skin tissues of HCs, DLE patients and SLE patients. We compared the difference in cell type composition, further described the cell subclusters and their functions, and finally investigated cell–cell interactions.

Based on scRNA-seq data analysis and immunofluorescent staining, we identified that there are more immune cells, such as T, B and NK

cells, in the cutaneous lesions of DLE than SLE. The B cell expansion in DLE was consistent with a previous study showing that B cells account for a greater proportion in DLE than in SLE[24]. These results suggested the distinct pathogenesis of DLE and SLE and provided a potential way to distinguish DLE and SLE. Furthermore, we identified the subclusters of some cell types and showed that the ISGs signature, autophagy signaling and neutrophil activation were enriched in DLE or SLE cells. Notably, we defined some epidermal and dermal cell subtypes with increased proportions/expressions in DLE and SLE, including CCL20$^+$ keratinocytes, CXCL1$^+$ fibroblasts, ISG$^{hi}$ CD4/CD8 T, ISG$^{hi}$ plasma, pDCs, and NK SCs. We also identified some important genes related to immune microenvironments in cutaneous lesions of lupus patients. For example, the G protein-coupled receptor GPR183, which has been reported to be involved in the migration and localization of multiple immune cells[54], was shown to be highly expressed in dermal T_SC0 in our dataset.

Combining DEGs and GO enrichment analysis, we found that some keratinocytes and fibroblasts in cutaneous lesions of lupus might function as immune cells. Diff era-4, Diff kera-5 and Diff kera-6 participated in some BPs, such as the regulation of innate immune response and antigen processing and presentation, and HLA$^+$Fib1 and CXCL1$^+$Fib were involved in the processes of leukocyte adhesion and activation[55]. HSP70-coding genes have been reported to assist in the delivery of antigens in both epidermis and dermis and to have an important function in immunoregulation of autoimmune diseases[56]. In this study, we found that HSP70-coding genes were highly expressed in epidermal T_SC2, B_SC0, M_SC2, and N_SC0 and dermal T_SC5, B_SC3, M_SC3, and N_SC1, which suggests that HSP70-coding genes may have an important function in cutaneous lesions of DLE and SLE.

Extensive evidence has shown the involvement of the interferon signaling pathway in the pathogenesis of lupus[28,57,58]. Our study identified some cell subtypes with high expression of ISGs, such as Diff kera-4, Pericyte2, epidermal T_SC1, dermal T_SC5, epidermal B_SC3, dermal B_SC4 and B_SC5, epidermal M_SC1, dermal M_SC6, epidermal N_SC2 and dermal N_SC1, which is consistent with a previous report that a strong interferon signature was identified in multiple cell subtypes of CLE[41]. Thus, studying the functions and mechanism of the activation of the interferon signaling pathway in lupus may help identify more intervention targets for lupus treatment.

Previous studies have shown that neutrophil dysfunction is related to the pathogenesis of SLE[59]. In this study, although we did not detect any neutrophil infiltration due to the preference error of the droplet-based 10× Genomic system, the neutrophil activation pathway was found in multiple cell types according to GO BP analysis. Furthermore, we identified neutrophils in cutaneous lesions of DLE and SLE by immunofluorescence staining for ELANE (a neutrophil elastase; Supplementary Fig. 8), suggesting that neutrophils are involved in the formation of cutaneous lesions of lupus. In addition, we found that the autophagy pathway, which has been reported to facilitate skin differentiation and epidermal proliferation and accelerate inflammation[60,61], was enriched in epidermal B cells of DLE and dermal Macro/DCs of SLE.

Cell communication analysis among different cell types identified the enhanced interactions of keratinocytes and Macro/DCs with other immune cells in the epidermis of DLE and SLE, which may contribute to an increased number and activation of immune cells in cutaneous lesions of DLE and SLE. Furthermore, ligand-receptor pairs, such as *CCL20_CXCR3*, *FAM3C_CLEC2D*, *FAM3C_HLA_C*, SPN_*ICAM1*, and *TNFSF9_TNFRSF9*, had higher mean expression levels in epidermal cells of DLE patients and SLE patients than in those of HCs. Similarly, higher interaction scores were found in endothelial cells, fibroblasts, and Macro/DCs in the dermis of DLE patients than in SLE patients and HCs, in which many receptors and ligands, such as *CCR7*, *CCR2*, *CXCR3*, and *CCL19*, were found to be highly expressed in DLE. Therefore, the potential cell-cell interactions found in our study will help understand

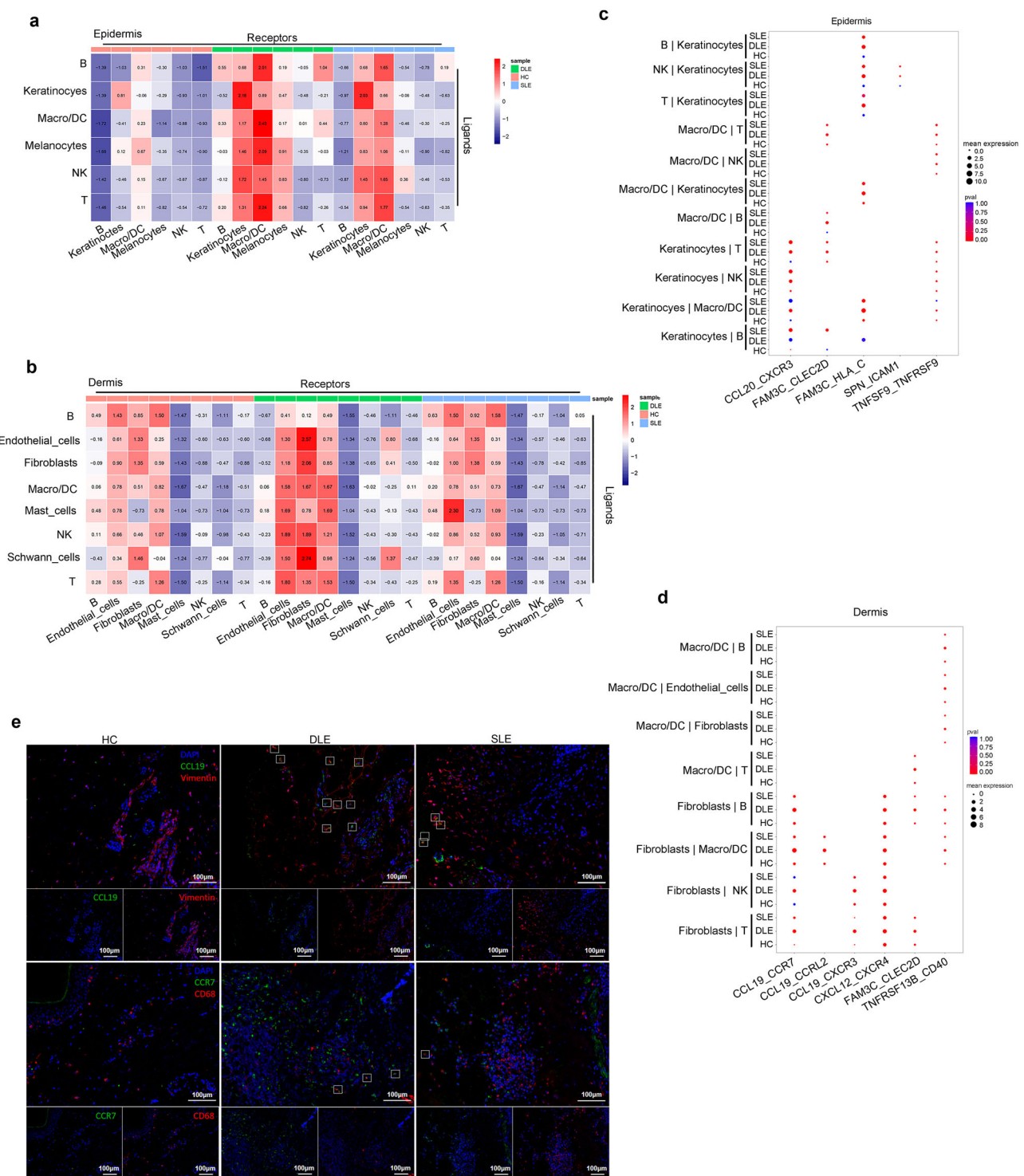

**Fig. 9 | Cell-cell interaction in epidermis and dermis. a, b** Heatmaps show interaction scores between cell types in epidermis (**a**) and dermis (**b**). The column annotation represents cells from HC/DLE/SLE. The purple color shows the low interaction scores, while the red color shows high interaction scores. **c, d** Bubble diagrams show the mean expression levels of selected ligand-receptor pairs in epidermal (**c**) or dermal (**d**) cell types of HC/DLE/SLE. The P values were calculated using one-side permutation test. **e** Represent images show immunofluorescence staining for CCL19 (green) expression in fibroblasts (Vimentin, red), CCR7 (green) expression in macrophages (CD68, red). Scale bar: 100 μM. Data are representative of 3 independent experiments. Macro/DC: macrophage/dendritic cell. Source data are provided as a Source Data file.

the changes in the immune microenvironment and cutaneous lesion formation in DLE and SLE.

While this study discovered critical transcriptional features of cells in cutaneous lesions of DLE and SLE, there were several limitations. We admit that because of mechanical separation and

sequencing data calculation deviation, we could not completely distinguish the cell types in the epidermis and dermis and had to remove the overlapping cells in the clustering analysis of the epidermis and dermis and in the sub-clustering analysis of cell types. Transcriptome analysis has a limited ability to identify immune cell

subclusters, and the use of some protein markers will help to identify cell types more accurately.

In summary, we have shown that the use of scRNA-seq by 10× Genomics is feasible and informative in the study of lupus, especially in distinguishing different cutaneous lesions of lupus patients. Our findings provide much information about DLE- and SLE-specific cellular and molecular signatures in cutaneous lesions compared to that of HCs, and we described in detail the similarities and differences in cell compositions between DLE and SLE, which will help understand the pathogenesis of DLE and SLE and develop novel and precise therapeutic targets for lupus erythematosus.

## Methods

### Clinical sample collection
All skin biopsies were collected at the dermatology biopsy center in the Second Xiangya Hospital of Central South University, Xiangya Hospital of Central South and University and Institute of Dermatology of Chinese Academy of Medical Sciences and Peking Union Medical College. Our study was approved by the Ethics Committee of the Second Xiangya Hospital, Xiangya Hospital and University and Institute of Dermatology of Chinese Academy of Medical Sciences and Peking Union Medical College. All patients and healthy controls involved in this study signed informed consent forms. Skin biopsy tissue samples (4 × 4 mm) were obtained from patients with clinically indicated LE by the classification criteria for SLE (1997/2012)[62] and CLE[63]. The average tissue mass of cutaneous lesions was 7 mg (3–10 mg).

### Preparation of single-cell suspensions
Once the pathological cutaneous lesion samples were removed from the patients, we immediately immersed them in tissue separation solution consisting of 10 mg dispase enzyme (Sigma, D4693-1G), 1 mL sterile PBS, and 2 mL sterile 1640 medium or physiological saline. Then, the biopsy tissue samples were transported to the laboratory within 30 minutes. The cutaneous lesion sample was placed in a petri dish with the epidermis facing up and the dermis facing down at 4 °C overnight. Skin biopsy tissue sample isolation and single-cell suspension preparation were performed the next day. We used an enzyme isolation method to make single-cell suspensions as previously described[64]. In short, enzyme reagents in a whole-skin dissociation kit (Miltenyi Biotec, 130-101-540) were mixed into EP tubes at a certain ratio according to the operation manual. The dermis and epidermis were separated by sterile tweezers and scissors and transferred into mixed enzyme solutions. Both the dermis and epidermis were subjected to a water bath at 37 °C for 3–4 hours. After the enzymatic reaction, the tissue residue solution was filtered by a strainer, and the filtrate was centrifuged at $300 \times g$ for 10 min at 4 °C. Furthermore, we separately added 100 μL magnetic beads to cells of the epidermis and dermis, incubated them at room temperature for 15 min, and then filtered them on a magnetic stand. The filtrate was collected, centrifuged at $300 \times g$ for 10 min at 4 °C, and then resuspended in 10% FBS. Ten microliters of the cell suspension were mixed with the same volume of Trypan blue for counting. The loading volume was verified by the cell concentration.

### scRNA-seq
scRNA-seq mainly includes GEM (gel bead-in-emulsion) generation, barcoding, cDNA amplification, library construction and sequencing. These steps were completed according to the user's instructions of Chromium Single Cell 3′ Reagent Kits v3.1 (10× Genomics, product code: 1000268, 1000215, 1000120) (https://www.10xgenomics.com/support/single-cell-gene-expression). Libraries were sequenced by an Illumina NovaSeq6000 System. Approximately 10,000 cells (targeting 5000–12,000) per sample were used for single-cell RNA sequencing.

### scRNA-seq data processing
Raw sequencing data were first filtered and then mapped to GRCh38 by using Cell Ranger Version 3.0.0 (https://www.10xgenomics.com). After the Cell Ranger indicator evaluation, cell doublets were removed by Scrublet[65]. Then, the Seurat v4 package[66] was used as the main tool for single-cell RNA sequencing analysis. Quality control was performed to remove low-quality cells with nFeature_RNA values less than 200 or greater than 5000, with percent.mt values over 20% and with percent.redcell values over 10%. The filtered gene expression matrix was normalized and scaled by a scaling factor of 10,000. Then, the variable genes were generated by the function FindVariableFeatures. Furthermore, the first 2,000 highly variable genes were selected for principal component analysis (PCA). The batch effect among different patients was removed by the Harmony package[67] (https://github.com/immunogenomics/harmony). Then, the ElbowPlot function was performed to identify meaningful PCs. The first 20–40 PCs were used for t-SNE (t-distributed stochastic neighbor embedding) clustering analysis. The DEGs of each cluster were determined by the FindAllMarkers function, which compared one cluster with all other cells. The function AverageExpression was applied to calculate the average gene expression of each cluster or cell type. The distribution of single cells in clusters, sample groups or each sample was visualized by t-SNE plots. Some specific gene expression levels were plotted by feature plots and stacked violin plots. Other R packages, such as pheatmap, ggplot2, dplyr and RColorBrewer, were used for stacked bar plots and heatmaps. We followed the same steps as above for the sub-clustering analysis. Due to sequencing bias, there were some other cells involved in the second clustering analysis of a cell type. To eliminate their interference, we removed these cells before sub-clustering analysis.

### Pseudotime trajectory analysis
Monocle2[68] was adopted to infer potential differentiation associations between some expanded subclusters of cell types. The genes used for pseudotime trajectory analysis came from the FindAllMarkers function in the Seurat package based on the Wilcoxon rank sum test and were filtered by pct.1 > 0.5 and pct.2 < 0.5. The newCellDataSet function in Monocle2 was applied to generate a data structure for trajectory analysis. Then, after normalization, reduceDimension and orderCells were used to reduce the dimensionality and order cells in the pseudotime trajectory.

### GO biological process analysis
The clusterProfiler package was used for the GO functional analysis of biological processes as previously described[69]. Briefly, the marker gene list of subclusters or samples was provided for ID conversion from gene symbols to Entrez ID, and the enriched GO functions were used for GO biological process analysis with a $p$ value <0.01 and a $q$-value <0.05. The bubble chart in this study visualizes the partially enriched biological processes of subclusters or samples.

### Similarity scores
The similarity scores were calculated based on the differentially expressed genes of each subcluster in our study and subclusters (SCs) reported in PBMCs of lupus[9]. We adopted the intersecting function of R to find the common number of differentially expressed genes between the subclusters in our study and SCs in PBMCs, and then we divided by the sum of two differentially expressed genes. The value obtained was the similarity score. The pheatmap package was used for visualization.

### Immunofluorescence
T cells, B cells and NK cells were separately stained for CD3 (MAB-0740, MXB Biotech), CD19 (1:1000, ab134114, Abcam) and CD56 (1:400, ab75813, Abcam) expression. CCL20+ keratinocytes were

stained with CCL20 (1:50, ab224188, Abcam) and KRT10 (1:500, ab76318, Abcam). CXCL1+ fibroblasts and HLA+ Fib1 were identified by staining CXCL1 (1:100, ab89318, Abcam) and HLA-DRB1 (1:1000, ab133578, Abcam) with Vimentin (1:20000, ab92547, Abcam). ELANE (1:2000, ELA-2, ab131260, Abcam) was used to stain neutrophils. CCL19 (1:500, 13397-1-AP, ProteinTech) staining was performed in fibroblasts, and CCR7 (1:500, 25898-1-AP, ProteinTech) staining was performed in macrophages (CD68, 1:400, ab955, Abcam). The second antibodies used in this study came from Abcam (Donkey Anti-Rabbit IgG, 1:1000, ab205722, Abcam). All immunofluorescence experiments were conducted as follows[70]. Briefly, skin biopsy samples were embedded in paraffin and sectioned into 4 μm thick sections. After the paraffin tissue sections were baked, dewaxed, and rehydrated, they were used for antigen retrieval under acidic conditions in a pressure cooker. After peroxidase enzymes were removed, the cutaneous lesions were sealed with 5% serum (Well-Biology, CN). Then, the samples were incubated with primary antibodies at room temperature for 1 h or at 4 °C overnight. After incubation with secondary antibodies and staining with reagents from an Opal 7-color IHC detection kit (NEL811001KT, PerkinElmer, Hopkinton, MA), histologic sections were observed with an LSM 510 confocal microscopy system (Leica) or PerkinElmer Vectra (Hopkinton, MA). All images were analyzed using inForm software compatible with the PerkinElmer Vectra.

## Cell-cell communication analysis and interaction scores

Cell-to-cell communication analysis of potential receptor-ligand pairings was conducted using CellPhoneDB[52]. For calculation of the average expression and product value of ligands and receptors, each cell type was labeled and randomly arranged 1000 times. To identify the meaningful interaction pairs, we calculated the p value and mean expression, and then we eliminated the pairs with a cutoff $p$ value lower than 0.05. The visualizations were performed in pheatmap and ggplot2. Furthermore, we counted the numbers of ligand-receptor pairs enriched between cell types, and the interaction scores were generated by the pheatmap function with the parameter scale = 'row' and clustering_distance_rows = " correlation" (Pearson correlation).

## Statistical analysis

The data are shown as the mean ± SD, and the statistical significance of the proportional differences in cell types among HC, DLE, and SLE was determined by one-way ANOVA test with Bonferroni's multiple comparisons test. Other biologically significant differences were determined by the default algorithm of R packages.

## Reporting summary

Further information on research design is available in the Nature Portfolio Reporting Summary linked to this article.

## Data availability

The processed data of this study have been deposited in the GEO database under accession code GSE179633. These data generated in this study are available within the Article, Supplementary Information or from the corresponding author upon reasonable request. Source data are provided with this paper.

## Code availability

All the codes used for processing and analyzing the data in this study have been deposited in an available GitHub repository[71] (https://doi.org/10.5281/zenodo.7193545).

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

## Acknowledgements

We thank the dermatology biopsy centers in the second Xiangya Hospital of Central South University, Xiangya Hospital of Central South University and Institute of Dermatology of Chinese Academy of Medical Sciences and Peking Union Medical College for providing the skin biopsies for this study. This work was supported by the National Natural Science Foundation of China (No. 82030097 and No. 81874243, Ming Zhao), CAMS Innovation Fund for Medical Sciences (CIFMS) (2019-I2M-5-033, M.Z.), the Key project for international and regional cooperation in science and technology innovation of Hunan province (2019WK2081, M.Z.), and the Project for leading talents in science and technology in Hunan province (2019RS3003, Ming Zhao).

## Author contributions

M. Zheng processed the data, performed the analysis, and wrote the manuscript. X.M., L.O., and Y.S. conducted the immunofluorescence staining of skin tissues. M.Zhao and Z.H. helped M. Zheng to analyze the subclusters. W.Z. and H.J. helped M. Zheng to collect and handle the clinical information of patients. Z.H., Y.K., and R.W. performed the treatment of single suspensions. Q.L., H.L., S.R., and W.S. performed the diagnosis of DLE and SLE and collected the skin tissues. S.L., H.W. and S.J. help collect samples of DLE and SLE and immunofluorescence staining of skin tissues. M. Zhao conceptualized and supervised this study and revised the manuscript. Q.L. conceptualized and supervised this study and revised the manuscript. All authors reviewed and approved the manuscript.

## Competing interests

The authors declare no competing interests.
