## [Peer Review File · Nature Communications]

Single-cell sequencing shows cellular heterogeneity of cutaneous lesions in lupus erythematosusREVIEWER COMMENTS

Reviewer #1 (Remarks to the Author):

This is a very good manuscript that reveals several interesting features of cells in the dermis and epidermis in both SLE and discoid LE. Both diseases have cutaneous lesions, but they are known to differ, however no study had looked before at the comparison. This paper is good, quite well written and the descriptions are clear. As every single cell work, it is very descriptive.

Some interesting findings are cells other than pDCs producing IFN, expansion of fibroblasts, infiltration of neutrophils, as well as infiltrating T cells also with an IFN response (ISG+GPR183+HSP α). Also, the pseudotime analysis is interesting and shows the trajectories of the various cells in the tissue. Just some minor comments:

1. When describing the several pathways found using GO, please do not write "and so on". This is not serious. If you do not want to describe all the pathways found, show a table.
2. Are the stainings done to verify some of the genes and cell types done on the same samples as those used for the scRNASeq?
3. The marker LC3B is indeed a marker of autophagy, but when cleaved is processed. I believe that the autophagy part is too speculative as it cannot be shown that indeed this is ongoing. The mere expression of the gene is not enough. It is already known that LC3B is expressed differently in various cell types.
3. On the expression of the MHC genes, if it is HLA-DP, please mention it as such. Or write HLA class II.
4. Line 182: typo: epidermis
5. Lines 80-84: I believe it is unnecessary to write methods in the introduction. I would take away from "We received a total of ...cell subset". All this is well described in the methods section. Finally, as a comment I find very interesting that ABC B cells are found in the dermis and epidermis as well.

Reviewer #2 (Remarks to the Author):

Overall comments

This is an interesting study that brings provocative data that could potentially shed light on the pathogenesis of two different types of lupus skin involvement. The authors analyzed scRNA-Seq data from epidermis and dermis cells in SLE and DLE patients and compared the data with a public dataset of full thickness skin biopsies from 3 healthy donors. They also use limited immunohistochemistry to validate some of their transcriptional results. The authors first analyzed major skin cell clusters from total epidermis or dermis, followed by breakdown and sub-cluster analysis of each cell type. The overall methodology for scRNA-Seq data processing and integration to detect major skin cell types looks sound, yet details about sub-cluster analysis are insufficient. The authors provide additional analysis investigating cell trajectories and predictions of cell-cell interactions.

Major comments:

1. The patient clinical and laboratory information provided is suboptimal. For example, there are validated measures of lupus skin disease activity (mCLASI) that would be very important to incorporate into the analysis. Information on the location of the rash/biopsies would also be desirable. This is particularly important as it has been shown that skin regions exposed to UV light display a different transcriptional profile compared to the regions that are not (doi: 10.1038/s42003-020-0922-4). Overall, the display of the currently included clinical and laboratory values should be improved.
2. While the approach to compare DLE and SLE lesions at the dermal and epidermal levels is appropriate, the choice of healthy full-thickness skin from the literature as reference control brings complexity and opens many questions regarding overall data interpretation. To start with, the

numbers of the cells loaded in the 10X are significantly different across HD, DLE and SLE (Table S2), plus the authors should consider artifacts due to methodological differences in sample processing between their samples and those from the literature.

3. The selection of only two DLE and SLE samples, respectively, for studies of the epidermis also decreases enthusiasm for the current version of this work, especially as more differences are found in the study of this skin layer cell composition/transcriptional profile, which has very low statistical power, compared with the larger dermis study that includes 5 and 7 DLE and SLE samples, respectively.

4. For sub-clustering analysis, using keratinocytes as an example, the authors integrated epidermal cells from two DLE and two SLE patients, plus 3 full-thickness healthy skin from public data and identified 15 sub-clusters in keratinocytes. It is not clear either why they chose to include in their analysis only 3 out of the 5 healthy skin samples from the public dataset. Based on supplementary figure S3, it seems like the samples from HD, DLE, SLE do not mix well, which could be caused by two obvious reasons: (a) the diversity clustering penalty parameter (θ) of the Harmony integration algorithm was not set to a sufficient level to mix healthy keratinocytes cells with keratinocytes in DLE/SLE patients; (b) Harmony is working fine and the sub-clusters do reflect biologically distinct keratinocyte subtypes. Using healthy control samples processed in the same way as the patient's and increasing the number of dermal DLE and SLE samples would be important to bring sound conclusions to these aspects of the study.

5. Interpreting cell proportions in this study is tricky, as the denominator can vary significantly between conditions (Healthy vs. DLE vs. SLE). This is not a big issue for proportion plots in Fig. 2a or b, because every major cell type is compared against total epidermis/dermis cells. It becomes more complicated in sub-cluster analysis, because for each cell type, its proportion in the total epidermis/dermis can vary drastically. Taking keratinocytes again as an example: according to Fig. 2, proportion for keratinocytes is ~30% and ~75% in epidermis for DLS and SLE, respectively. From Fig. S1, we can see from the public healthy full-thickness skin, the proportion of keratinocytes ranges from ~25% to ~45%. This means the proportion of keratinocytes in the total skin cells can vary significantly between conditions. Did the authors consider this factor when plotting and interpreting the proportion bar plot for sub-cluster analysis, such as Fig. 3b and other similar plots? As in the comment above, using similarly processed healthy control samples as well as increasing the number of lupus patient samples would enable to get more sound conclusions.

6. Although the authors already did QC and percent.mt filtering and doublet removal at the beginning, it is still worth checking whether in the sub-clustering analysis, any sub-cluster displays high mito percentage, low RNA counts, as well as evaluate the possibility of leftover doublet clusters by checking known markers on all sub-clusters.

7. The authors need to improve the figures and their resolution (example: Fig 3 d, gene name hard to read). The colors for DLE and SLE used in Fig 2C have not accurately described in the figure annotation. Some figure legends do not align with the figures (for example FigS2). Fig. 6l seems to be a repeat of 6D rather than the display of dermal B cell subcluster data. Figures 9 A and D are unreadable. Also, supplement table S1 was split across several pages, which is difficult for readers to follow.

Reviewer #3 (Remarks to the Author):

Zheng et al show for the first time the characterization of the cell types present in the skin of patients with DLE and SLE, at the single cell RNA sequencing level. This adds valuable information to our knowledge about both which cell types are present, and the differences between the two manifestations of lupus lesions.

That being said, I have some major and minor reservations about this article.

Major points:

1. General interest of reading style. Considering the paper is largely built of same technique used 6 times for 6 different cells types (keratinocytes, fibroblasts, T cells, B cells, DCs and NK cells), the manuscript reads slightly monotonously in my opinion. Although I can appreciate how thorough this is, it is difficult to hold attention. The flow of the article is also sometimes jumpy, for example sudden mention of the potential involvement of anti-virus response in LE lesions (line 216), which seems to come from nowhere.
2. N numbers – only 2 patients in epidermal group for both DLE and SLE were analyzed. I really doubt if you make the conclusions in this article based on this number of patients? Table S1 with the clinical information is also not readable at all.
3. Also use of 'SC' for clusters of all cell types analyzed, no matter whether keratinocyte, fibroblast, T cell, B cell, DC or NK cells makes reading difficult? Not sure how to remedy this, you need to call them something, and there are lots of them?
4. Over interpretation, for example: Conclusion strong? Line 182 – found to be involved in local infiltration of immune cells – this has not been demonstrated by the authors. Only that they expressed genes suggesting this. Also – infiltration is not physically close to these cells, or not?
5. Issues with Figure 9. Fig 9 in my opinion is the most interesting, where the authors calculate likely interactions between cell groups based on expression of ligands and receptors. Although pleasing to look at, I don't feel like Fig 9a,b,e and f add much to the story. The information about strength of interaction is better presented in Fig. 9 parts c,d,g,and h. Most critically, none of this hypothetical and most interesting analysis is validated in the tissue. I find this odd, considering the authors have validated findings in earlier figures.

Minor points and further thoughts:

1. A study was published recently suggesting that the stem cells and transient amplifying cells of the skin in SLE patients may be senescent, and partly responsible for scar formation in some SLE lesions (PMID: 34273349). Did the authors considering looking at a stem cell phenotype, or indeed markers of senescence such as p16 or p21?
2. Many small typos. Figures mis-numbered. Please check all. Fig S1a not referenced in manuscript. Fig S2 legend does not read easily. CD1c in some places should probably read CD11c? Types: Line 180 'type interferon'. Line 182 – 'eoidermisof'.
3. Line 168 = autophagy involved in infiltration of immune cells and local inflammation? Autophagy of which cells? I don't understand which cells this comment is in relation to.
4. What is the age of healthy control patients in database downloaded for controls? Missing from Table S1.
5. SC7 proliferating keratinocyte sub group – did the authors also look at expression of classical markers of proliferation in this cluster, for example Ki67 or PCNA?
6. Pseudotime trajectory Fig. 3g – very minimal information – can the authors put more labels in at least?
7. Are these skin lesions with scar formation? Would be interesting to know, because of this link with activated types of fibroblasts.
8. Also how many tissues were stained for these validation experiments in Figures 2-4, and why was this not performed in the other cell subsets (T cells, B cells, DCs, NK cells)?

Point-by-Point Answers to the Reviewers' Comments

We deeply appreciate you and the reviewers for their encouraging and insightful
suggestions and comments on improving our manuscript. As requested, we have
performed additional experiments as described in detail below and added the
explanation for some questions. All the changes have been highlighted in red font in
the revised manuscript. With these extensive revisions, we believe that our manuscript
has been significantly improved.

**Reviewer 1:**

*This is a very good manuscript that reveals several interesting features of cells in the*
*dermis and epidermis in both SLE and discoid LE. Both diseases have cutaneous lesions,*
*but they are known to differ, however no study had looked before at the comparison.*

*This paper is good, quite well written and the descriptions are clear. As every single*
*cell work, it is very descriptive.*

*Some interesting findings are cells other than pDCs producing IFN, expansion of*
*fibroblasts, infiltration of neutrophils, as well as infiltrating T cells also with an IFN*
*response (ISG+GPR183+HSP α). Also, the pseudotime analysis is interesting and*
*shows the trajectories of the various cells in the tissue.*

*1. when describing the several pathways found using GO, please do not write "and so*
*on". This is not serious. If you do not want to describe all the pathways found, show a*
*table.*

**Response:** We deeply appreciate the reviewer's comment and suggestion. We deleted
"and so on" in the revised manuscript and showed all pathways enriched by GO in
supplementary Table S6 and Table S8.

2. *Are the stainings done to verify some of the genes and cell types done on the same*
*samples as those used for the scRNASeq?*

**Response:** We thank for the reviewer's comment. The staining experiments for
verification of genes and cell types were done on both the same samples as those used
for the scRNA-seq and the samples from other HC, DLE and SLE patients.

3. *The marker LC3B is indeed a marker of autophagy, but when cleaved r processed. I*
*believe that the autophagy part is too speculative as it cannot be shown that indeed this*
*is ongoing. The mere expression of the gene is not enough. It is already known that*
*LC3B is expressed differently in various cell types.*

**Response:** We are grateful to the reviewer for this insightful comment. We agree with
the reviewer that the only LC3B marker cannot reflect the autophagy of fibroblast in
skin lesions of DLE and SLE. Moreover, we added the samples of healthy subjects,
DLE patients, and SLE patients in the revised manuscript. According to the new data
analysis, we found that "autophagy" was only enriched in B cells from epidermal
samples of DLE patients (revised Fig 6e) and Macro/DC cells from dermal samples of
SLE patients (revised Fig7j), but not in fibroblast from DLE and SLE patients.
Therefore, we removed the LC3B staining result in the revised manuscript.

4. *On the expression of the MHC genes, if it is HLADR, please mention it as such. or*
*write HLA class II.*

**Response:** We thank reviewer's suggestion. We changed "the MHC genes" to "HLADR"
in the revised manuscript (Page 8 and Page10).

5. *Line 182: typo: epidermis*

**Response:** We sorry for the mistake. We have corrected the typo in the revised
manuscript.

*6. Lines 80-84: I believe it is unnecessary to write methods in the introduction. I would*
*take away from "We received a total ofcell subset". All this is well described in the*
*methods section.*

**Response:** We agree with the reviewer. According to the suggestion, we removed the
methods from introduction (Page 4).

**Reviewer 2:**

*This is an interesting study that brings provocative data that could potentially shed light*
*on the pathogenesis of two different types of lupus skin involvement. The authors*
*analyzed scRNA-Seq data from epidermis and dermis cells in SLE and DLE patients*
*and compared the data with a public dataset of full thickness skin biopsies from 3*
*healthy donors. They also use limited immunohistochemistry to validate some of their*
*transcriptional results. The authors first analyzed major skin cell clusters from total*
*epidermis or dermis, followed by breakdown and sub-cluster analysis of each cell type.*
*The overall methodology for scRNA-Seq data processing and integration to detect*
*major skin cell types looks sound, yet details about sub-cluster analysis are insufficient.*
*The authors provide additional analysis investigating cell trajectories and predictions*
*of cell-cell interactions.*

*1. The patient clinical and laboratory information provided is suboptimal. For example,*
*there are validated measures of lupus skin disease activity (mCLASI) that would be very*
*important to incorporate into the analysis. Information on the location of the*
*rash/biopsies would also be desirable. This is particularly important as it has been*

*shown that skin regions exposed to UV light display a different transcriptional profile*
*compared to the regions that are not (doi: 10.1038/s42003-020-0922-4). Overall, the*
*display of the currently included clinical and laboratory values should be improved.*

**Response:** We deeply appreciate the reviewer's comment and suggestion. We are sorry
for the suboptimal clinical and laboratory information of patients. The information
including mCLASI, location of biopsies, etc has been shown in the revised table S1.

*2. While the approach to compare DLE and SLE lesions at the dermal and epidermal*
*levels is appropriate, the choice of healthy full-thickness skin from the literature as*
*reference control brings complexity and opens many questions regarding overall data*
*interpretation. To start with, the numbers of the cells loaded in the 10X are significantly*
*different across HD, DLE and SLE (Table S2), plus the authors should consider artifacts*
*due to methodological differences in sample processing between their samples and*
*those from the literature.*

**Response:** We agree with the reviewer. To eliminate the differences in sample
processing and cell numbers, we collected normal skin tissues from 5 healthy subjects
and divided the epidermis and/or dermis to perform scRNA-seq in the same way as
experiments in DLE and SLE samples. Therefore, we showed the new results in the
revised manuscript.

*3. The selection of only two DLE and SLE samples, respectively, for studies of the*
*epidermis also decreases enthusiasm for the current version of this work, especially as*
*more differences are found in the study of this skin layer cell*
*composition/transcriptional profile, which has very low statistical power, compared*
*with the larger dermis study that includes 5 and 7 DLE and SLE samples, respectively.*

**Response:** We thank the reviewer's comment. As suggestion, we added the epidermis
samples from 3 DLE patients and 3 SLE patients. The new results of comparison in
epidermis were obtained from 5 DLE patients, 5 SLE patients and 4 healthy subjects in
the revised manuscript.

*4. For sub-clustering analysis, using keratinocytes as an example, the authors*
*integrated epidermal cells from two DLE and two SLE patients, plus 3 full-thickness*
*healthy skin from public data and identified 15 sub-clusters in keratinocytes. It is not*
*clear either why they chose to include in their analysis only 3 out of the 5 healthy skin*
*samples from the public dataset. Based on supplementary figure S3, it seems like the*
*samples from HD, DLE, SLE do not mix well, which could be caused by two obvious*
*reasons: (a) the diversity clustering penalty parameter (theta) of the Harmony*
*integration algorithm was not set to a sufficient level to mix healthy keratinocytes cells*
*with keratinocytes in DLE/SLE patients; (b) Harmony is working fine and the sub-*
*clusters do reflect biologically distinct keratinocyte subtypes. Using healthy control*
*samples processed in the same way as the patient's and increasing the number of*
*epidermal DLE and SLE samples would be important to bring sound conclusions to*
*these aspects of the study.*

**Response:** We appreciate the reviewer's suggestion. To eliminate the differences in
sample processing, we collected normal skin tissues from 5 healthy subjects and divided
the epidermis and dermis to perform scRNA-seq in the same way as experiments in
DLE and SLE samples. In the revised manuscript, we showed the new results compared
with our data in healthy subjects.

*5. Interpreting cell proportions in this study is tricky, as the denominator can vary*
*significantly between conditions (Healthy vs. DLE vs. SLE). This is not a big issue for*
*proportion plots in Fig. 2a or b, because every major cell type is compared against*

*total epidermis/dermis cells. It becomes more complicated in sub-cluster analysis,*
*because for each cell type, its proportion in the total epidermis/dermis can vary*
*drastically. Taking keratinocytes again as an example: according to Fig. 2, proportion*
*for keratinocytes is ~30% and ~75% in epidermis for DLE and SLE, respectively. From*
*Fig. S1, we can see from the public healthy full-thickness skin, the proportion of*
*keratinocytes ranges from ~25% to ~45%. This means the proportion of keratinocytes*
*in the total skin cells can vary significantly between conditions. Did the authors*
*consider this factor when plotting and interpreting the proportion bar plot for sub-*
*cluster analysis, such as Fig. 3b and other similar plots? As in the comment above,*
*using similarly processed healthy control samples as well as increasing the number of*
*lupus patient samples would enable to get more sound conclusions.*

**Response:** We appreciate the valuable comment of the reviewer. We agree with the
reviewer. The variations of the denominator such as keratinocytes proportions in DLE,
SLE and HC were not considered in our study, which affected the accuracy of sub-
cluster proportion analysis. As the proportions of each cell type are unlikely to be
consistent across DLE, SLE and HC, we changed the calculation method in the revised
manuscript. We used the total cell number of each subcluster as denominator, and then
calculate the proportion of each subcluster from DLE, SLE or HC. The results were
shown in the revised Fig.3b, Fig.4b, Fig.5b, Fig.5g, Fig.6b, Fig.6g, Fig.7b, Fig.7g,
Fig.8b, and Fig.8g. As suggested, we added 4 epidermal tissues and 4 dermal tissues of
healthy controls, and 3 epidermal tissues of DLE and 3 epidermal tissues of SLE in
order to get more sound conclusions.

*6. Although the authors already did QC and percent.mt filtering and doublet removal*
*at the beginning, it is still worth checking whether in the sub-clustering analysis, any*
*sub-cluster displays high mito percentage, low RNA counts, as well as evaluate the*
*possibility of leftover doublet clusters by checking known markers on all sub-clusters.*

**Response:** We thank for the suggestion of the reviewer. As suggested, we check the
quality of data for the sub-clustering analysis and did not find the high mito percentage
and low RNA counts in the sub-clustering analysis. Here, we take the epidermal T cell
sub-clustering results as an example to show the features (200-5000), mito percentage
(<20%) and the percent of red cells (<10%) in epidermal T cells performed sub-
clustering analysis (as shown below). In addition, we checked the known markers
expression on all sub-clusters. We did not find the leftover doublet cluster in most of
sub-clusters. However, we found that the tiny proportions of T cells, B cells and
Macro/DC subclusters in epidermis expressed the keratinocyte marker KRT1 or KRT14
(Fig S3a, Fig S4a, S4e, Fig S5a, S5f, Fig S6a, S6e, Fig S7a, S7e), which have no
obvious influence on our sub-clusters analysis of these immune cells in epidermis.

*7. The authors need to improve the figures and their resolution (example: Fig 3 d, gene*
*name hard to read). The colors for DLE and SLE used in Fig 2C have not accurately*
*described in the figure annotation. Some figure legends do not align with the figures*
*(for example FigS2). Fig. 6I seems to be a repeat of 6D rather than the display of dermal*
*B cell subcluster data. Figures 9 A and D are unreadable. Also, supplement table S1*
*was split across several pages, which is difficult for readers to follow.*

**Response:** We thank for the reviewer's thoughtful comment and are sorry for the

unsatisfactory figures. We have improved the figures and their resolutions according to
the reviewer's suggestion. In addition, we carefully checked and corrected these
mistakes in the figure legends. The revised table S1 was re-submitted in the
supplementary data.

**Reviewer 3:**

*Zheng et al show for the first time the characterization of the cell types present in the*
*skin of patients with DLE and SLE, at the single cell RNA sequencing level. This adds*
*valuable information to our knowledge about both which cell types are present, and the*
*differences between the two manifestations of lupus lesions. That being said, I have*
*some major and minor reservations about this article.*

*Major points:*

*1. General interest of reading style. Considering the paper is largely built of same*
*technique used 6 times for 6 different cells types (keratinocytes, fibroblasts, T cells, B*
*cells, DCs and NK cells), the manuscript reads slightly monotonously in my opinion.*
*Although I can appreciate how thorough this is, it is difficult to hold attention. The flow*
*of the article is also sometimes jumpy, for example sudden mention of the potential*
*involvement of anti-virus response in LE lesions (line 216), which seems to come from*
*nowhere.*

**Response:** We deeply appreciate the reviewer's insightful comment and suggestion. In
the manuscript, we displayed the cell atlas of epidermis and dermis in skin lesions of
DLE and SLE patients. We showed the cell subclusters, gene lists, and GO pathways of
6 different cell types, which will help us understand the pathogenesis of lupus cutaneous
lesion and the difference in cutaneous lesion between DLE and SLE. To avoid the
monotonous description in the results, we try our best to improve the writing style to
make it more attractive. And we also revised these sentences to make their presence

more reasonable.

*2. N numbers - only 2 patients in epidermal group for both DLE and SLE were analyzed.*

*I really doubt if you make the conclusions in this article based on this number of patients?*

*Table S1 with the clinical information is also not readable at all.*

**Response:** We agree with the reviewer. In order to enhance the reliability of the
conclusions, we added the epidermis samples from 3 DLE patients and 3 SLE patients.

The new results of comparison in epidermis were obtained from 5 DLE patients, 5 SLE
patients and 4 healthy subjects in the revised manuscript. We are sorry for the
unreadable table S1. We have revised the Table S1 and re-submitted it in the
supplementary data.

*3. Also use of 'SC' for clusters of all cell types analyzed, no matter whether keratinocyte,*
*fibroblast, T cell, B cell, DC or NK cells makes reading difficult? Not sure how to*
*remedy this, you need to call them something, and there are lots of them?*

**Response:** We appreciate the suggestion of the reviewer. We changed “SC” for clusters
of all cell types to “T_SC, B_SC, M_SC, N_SC” to represent the subclusters of T cell,
B cell, Macro/DC, and NK cells in the revised manuscript.

*4. Over interpretation, for example: Conclusion strong? Line 182 - found to be*
*involved in local infiltration of immune cells - this has not been demonstrated by the*
*authors. Only that they expressed genes suggesting this. Also - infiltration is not*
*physically close to these cells, or not?*

**Response:** We agree with the reviewer. We revised the overinterpretation of conclusion
on line 182 and other places in the revised manuscript.

*5. Issues with Figure 9. Fig 9 in my opinion is the most interesting, where the authors*
*calculate likely interactions between cell groups based on expression of ligands and*
*receptors. Although pleasing to look at, I don ' t feel like Fig 9a,b,e and f add much to*
*the story. The information about strength of interaction is better presented in Fig. 9*
*parts c,d,g,and h. Most critically, none of this hypothetical and most interesting analysis*
*is validated in the tissue. I find this odd, considering the authors have validated findings*
*in earlier figures.*

**Response:** We appreciated the reviewer's thoughtful comment. The analysis of
interactions between cell groups based on expression of ligands and receptors is one of
important parts in the manuscript. In order to make Fig. 9 present more information
about interactions, we replaced the Fig. 9a, b, e and f with bubble plots (Fig. 9c, 9d)
which showed different mean expression levels of several ligand-receptor pairs in DLE,
SLE and HC. The Fig. 9c, d, g, and h were improved to present the difference in strength
of cell interactions among DLE, SLE and HC (Fig. 9a and 9b). In addition, we also
performed the immune-staining of ligand and receptor in the skin tissues of DLE, SLE
and HC to validate our findings in cell interactions analysis (Fig. 9e).

**Minor points:**

*1. A study was published recently suggesting that the stem cells and transient amplifying*
*cells of the skin in SLE patients may be senescent, and partly responsible for scar*
*formation in some SLE lesions (PMID: 34273349). Did the authors considering looking*
*at a stem cell phenotype, or indeed markers of senescence such as p16 or p21?*

**Response:** We thank for the suggestion of reviewer. We checked the expression of
senescence markers p16 and p21 in our scRNA-seq data. We did not find high
expression of the CDKN2A (p16) in epidermal or dermal cells from HC, DLE and SLE.

Although did observe widely high expression of CDKN1A (p21) in epidermal and
dermal cells from HC, DLE and SLE, there is no significant difference in SLE and DLE
compared to HC (as shown below, purple indicates CDKN1A expression).

2. Many small typos. Figures mis-numbered. Please check all. Fig S1a not referenced
in manuscript. Fig S2 legend does not read easily. CD1c in some places should probably
read CD11c? Types: Line 180 'type interferon' . Line 182 - 'eoidermisof' .

**Response:** We are sorry for these typos in our manuscript and figures. We checked the
manuscript and figures carefully and corrected these mistakes in the revised manuscript
and figures. CD1C gene encodes a member of the CD1 family of transmembrane

glycoproteins, which is marker of classical DC cells. Therefore, we used CD1C and
other known markers to perform the subcluster analysis for Macro/DC in our study.

*3. Line 168 = autophagy involved in infiltration of immune cells and local inflammation?*
*Autophagy of which cells? I don ' t understand which cells this comment is in relation*
*to.*

**Response:** We thank for the reviewer's comment. In the revised manuscript, we added
the samples of DLE, SLE and HC for scRNA-seq. According to the new data analysis,
we found that autophagy was only enriched in epidermal B cells of DLE (Fig 6e) and
dermal Macro/DC of SLE (Fig7j). Therefore, we removed the description about
autophagy in keratinocyte.

*4. What is the age of healthy control patients in database downloaded for controls?*
*Missing from Table S1.*

**Response:** In the revised manuscript, we collected the skin samples of healthy controls
for scRNA-seq according to the reviewer's suggestion. The ages of healthy controls
were added in the revised Table S1.

*5. SC7 proliferating keratinocyte sub group - did the authors also look at expression*
*of classical markers of proliferation in this cluster, for example Ki67 or PCNA?*

**Response:** We thank the reviewer's suggestion. We found that the classical proliferation
markers such as MKI67(Ki67) and PCNA were highly expressed in the cycling
keratinocytes (Fig. 3c).

*6. Pseudotime trajectory Fig. 3g - very minimal information - can the authors put*

*more labels in at least?*

**Response:** We agree with the reviewer. We added the labels in pseudotime trajectory of
revised Fig. 3g.

*7. Are these skin lesions with scar formation? Would be interesting to know, because of*
*this link with activated types of fibroblasts.*

**Response:** We thank for the comment. These skin lesions from patients did not contain
scar formation, which was shown in Table S1.

*8. Also how many tissues were stained for these validation experiments in Figures 2-4,*
*and why was this not performed in the other cell subsets (T cells, B cells, DCs, NK*
*cells)?*

**Response:** We appreciate the comment of reviewer. Totally 6 DLE skin tissues, 13 SLE
skin tissues and 8 healthy control skin tissues were used for staining experiments in
Figures 2-4. We mainly validated the expression of some immune-related genes in
keratinocyte and fibroblast to indicate their non-classical functions in skin lesions of
DLE and SLE. However, as the cell subsets of immune cells such as T cells and B cells
have been identified previously by other studies, we did not validate the cell subsets of
immune cells in this study.

REVIEWER COMMENTS

Reviewer #1 made a comment to the editor that they were satisfied.

Reviewer #2 (Remarks to the Author):

The authors have addressed some of the reviewer's comments especially as it refers to the number of samples, which previously was insufficient to draw conclusions. They have also improved the quality of tables and samples.

There are many remaining issues, however, including the poor quality of the writing and the interpretation of some of the data. Indeed, there are statements either exaggerated or not supported by solid evidence throughout. For example, they use GO analyses to determine the differential function of cells in health and disease, and this, as raised by a previous reviewer in the context of autophagy, can be an overestimation. In addition, there is misinterpretation of the literature used to justify the functional classification of some of the cell types. For example, they identify B cell clusters over-expressing HSP transcripts and use this feature to classify the cells as "antigen-presenting cells", while the quoted paper that should support this assignation (ref. 32) describes the presence of anti-HSP autoantibodies in SLE sera and does not functionally characterize any cell type.

Classification of T and B cells is overall very poor. While lack of full definition of subsets could be acceptable considering the complexity of purifying tissue lymphocytes, the authors still claim to have identified some subsets (e.g., ABCs, DN2s) in spite of the fact that the markers that they depict do not fit the expression pattern of their corresponding sub-clusters (see Fig. 6. D and I, where bona fide DN2 markers such as TBX21, FGR and FCRLs are not expressed by one sub-cluster, but by different B cell clusters). On the other hand, recent papers characterizing CLE and SLE skin at the transcriptional level and describing expansions of B cells in CLE are not quoted (doi.org/10.3389/fimmu.2021.775353). In Fig. 7, myeloid cells co-expressing CD14 and CD16 are classified as macrophages. Recent reports on scRNAseq in lupus skin describe CD16+ myeloid cells as Dendritic cells, and so on. There are also problems with the provided patient's data in revised Suppl. Table 1. Thus, at least two SLE patients seem to have positive anti-dsDNA antibody titers while their ANAs are negative. Overall, the number of miss-interpreted and/or inconsistent results decreases enthusiasm for the paper in its current form.

Reviewer #3 (Remarks to the Author):

The authors have addressed my comments satisfactorily.

Point-by-Point Answers to the Reviewers' Comments

I would like to thank the reviewers for the favorable evaluation and
constructive comments concerning our manuscript. The manuscript has been revised
according to the reviewer's suggestions. Below are our point-to-point responses (Roman)
to the reviewers' critiques (Italic). Changes are highlighted (red font) in the
revised manuscript and online supplement.

*Reviewer #1 made a comment to the editor that they were satisfied.*

**Response:** We are very glad that the Reviewer 1 are satisfied with our reply.

*Review #2*

*The authors have addressed some of the reviewer's comments especially as it refers to*
*the number of samples, which previously was insufficient to draw conclusions. They*
*have also improved the quality of tables and samples.*

*1. There are many remaining issues, however, including the poor quality of the writing*
*and the interpretation of some of the data. Indeed, there are statements either*
*exaggerated or not supported by solid evidence throughout. For example, they use GO*
*analyses to determine the differential function of cells in health and disease, and this,*
*as raised by a previous reviewer in the context of autophagy, can be an overestimation.*

**Response:** We deeply appreciate the reviewer's comment and suggestion. We feel sorry
for poor quality of the writing and improper interpretation of some of the data. First,
we tried our best to improve the writing of our manuscript and asked professional native
English editor to help revise our manuscript. Second, we also removed or amended
some statements in the revised manuscript. For example,

(1) We removed this sentence “The results suggested that fibroblast might transform
into the inflammatory fibroblast involved in immune cell migration, activation, and
inflammatory response in cutaneous lesions of DLE and SLE” in page 11.

(2) We changed “Taken together, epidermal and dermal B cells including ISG^{hi} plasma
cells and HSP^{hi} B cells were expanded in the epidermis and dermis of DLE and SLE,
especially in dermis of DLE, which may play an important role in the cutaneous lesions
of lupus” to “Taken together, epidermal and dermal B cells including ISG^{hi} plasma cells
and HSP^{hi} B cells were expanded in the epidermis and dermis of DLE and SLE,
especially in dermis of DLE” in Page 16.

(3) We changed “In addition, we also found that the autophagy pathway, which has been
reported to facilitate skin differentiation and epidermal proliferation and accelerate
inflammation, was enriched in epidermal B cells of DLE and dermal Macro/DCs of
SLE, which suggested that autophagy may be related with innate and adaptive immune
responses in cutaneous lesions of lupus” to “In addition, we also found that the
autophagy pathway, which has been reported to facilitate skin differentiation and
epidermal proliferation and accelerate inflammation, was enriched in epidermal B cells
of DLE and dermal Macro/DCs of SLE” in page 22-23.

*2. In addition, there is misinterpretation of the literature used to justify the functional*
*classification of some of the cell types. For example, they identify B cell clusters over-*
*expressing HSP transcripts and use this feature to classify the cells as “antigen-*
*presenting cells ” , while the quoted paper that should support this assignation (ref. 32)*
*describes the presence of anti-HSP autoantibodies in SLE sera and does not*
*functionally characterize any cell type.*

**Response:** We thank the thoughtful suggestion of the reviewer. We are sorry for the
misusing of literature to prove HSP^{hi} B cells’ s functions in antigen presentation. To
support our assignation of “antigen-presenting cells”, we changed ref 32 to the correct

references (ref 43, 44, page 15), which described that heat shocked proteins play a role
in the delivery and presentation of antigenic peptides.

*3. Classification of T and B cells is overall very poor. While lack of full definition of*
*subsets could be acceptable considering the complexity of purifying tissue lymphocytes,*
*the authors still claim to have identify some subsets (e.g., ABCs, DN2s) in spite of the*
*fact that the markers that they depict do not fit the expression pattern of their*
*corresponding sub-clusters (see Fig. 6. D and I, where bona fide DN2 markers such as*
*TBX21, FGR and FCRLs are not expressed by one sub-cluster, but by different B cell*
*clusters).*

**Response:** We agree with the reviewer. Due to the complexity of purifying tissue
lymphocytes, it is difficult to fully definite the subsets of lymphocytes. As shown in the
revised Fig 6d, 6i, Fig S5f and S5l, the marker genes of some cell subsets such as ABCs
(ITGAX, TBX21, FCRL2¹, classical marker genes of ABCs) did not express in one
sub-cluster, so we cannot define the subset as ABCs. In the revised manuscript, we
removed these incorrect definitions of cell subsets (Fig 6d, 6h, Fig S5f, S5l, Page15).

*4. On the other hand, recent papers characterizing CLE and SLE skin at the*
*transcriptional level and describing expansions of B cells in CLE are not quoted*
*(doi.org/10.3389/fimmu.2021.775353).*

**Response:** We thank for the reviewer's suggestion. The recent paper characterizing
CLE and SLE skin at the transcriptional level and describing expansions of B cells in
CLE, which provides important support for the findings of our manuscript. We quote
the paper in the revised manuscript (ref 24, page 6, 14, 21).

*5. In Fig. 7, myeloid cells co-expressing CD14 and CD16 are classified as*

*macrophages. Recent reports on scRNAseq in lupus skin describe CD16+ myeloid*
*cells as Dendritic cells, and so on.*

**Response:** We appreciate the reviewer's suggestion. According to the recent report on
scRNAseq in lupus skin, myeloid cells co-expressing CD14 and CD16 should not be
classified as macrophages. In the revised manuscript, we adopted the markers from the
recent report, by which they identified CD16⁺ myeloid cells as CD16⁺ Dendritic cells²,
and other markers for macrophages^{3,4} to define the classification of myeloid cells. The
results showed that epidermal M_SC0 and M_SC1 highly expressed marker genes of
CD16⁺DC (FCGR3A, CXCL10, CXCL11) and macrophages (TREM2, MRC1, CD68,
CD163), thus we annotated these two epidermal Macro/DC subclusters as the mix
subcluster of macrophages and DCs (Fig 7d). As to dermal Macro/DC subclusters, we
found that dermal M_SC6 highly expressed marker genes of CD16⁺DC (FCGR3A,
CXCL10, CXCL11), which was identified as DCs. Dermal M_SC2 expressed marker
genes of DC (CLEC10A and CD1C), as well as marker genes of macrophage (TREM2,
MRC1, CD163 and CD68), which was described as the mix subcluster of macrophages
and DCs (Fig 7i).

*6. There are also problems with the provided patient's data in revised Suppl. Table 1.*
*Thus, at least two SLE patients seem to have positive anti-dsDNA antibody titers*
*while their ANAs are negative.*

**Response:** We appreciate the remind of reviewer. We are very sorry that we missed the
ANA titers of several patients in the Suppl. Table 1. We rechecked carefully the clinical
information of the three patients without ANA titers shown in the Suppl. Table 1 and
found that two of them had the positive ANA antibody test result, which have been
added in the revised Suppl. Table 1. Another SLE patient did not test the ANA antibody
when we collected her skin sample, however she had previously been diagnosed as SLE

based on the Systemic Lupus International Collaborating Clinics (SLICC) 2012
classification criteria.

1 Dominguez Conde, C. *et al.* Cross-tissue immune cell analysis reveals tissue-
specific features in humans. *Science* **376**, eab15197,
doi:10.1126/science.ab15197 (2022).

2 Billi, A. C. *et al.* Nonlesional lupus skin contributes to inflammatory education
of myeloid cells and primes for cutaneous inflammation. *Sci. Transl. Med.* **14**,
eabn2263, doi:10.1126/scitranslmed.abn2263 (2022).

3 Zhang, L. *et al.* Single-Cell Analyses Inform Mechanisms of Myeloid-
Targeted Therapies in Colon Cancer. *Cell* **181**, 442-459 e429,
doi:10.1016/j.cell.2020.03.048 (2020).

4 Xue, D., Tabib, T., Morse, C. & Lafyatis, R. Transcriptome landscape of
myeloid cells in human skin reveals diversity, rare populations and putative
DC progenitors. *J. Dermatol. Sci.* **97**, 41-49,
doi:10.1016/j.jdermsci.2019.11.012 (2020).

*Reviewer #3 (Remarks to the Author):*

*The authors have addressed my comments satisfactorily.*

**Response:** We are very glad that the Reviewer 3 are satisfied with our reply.

REVIEWERS' COMMENTS

Reviewer #2 (Remarks to the Author):

The authors have addressed my latest comments and I would now recommend this paper for publication.

Point-by-Point Answers to the Reviewers' Comments

We sincerely thank all reviewers for their valuable feedback that help us to improve our manuscript. The reviewer comments are laid out below in italicized font. Our response is given in Roman font and changes to the manuscript are given in red text.

Reviewer #2 (Remarks to the Author):

The authors have addressed my latest comments and I would now recommend this paper for publication.

Response: We are delighted that the Reviewer 2 is satisfied with our reply.